



# Melt pond fractions on Arctic summer sea ice retrieved from Sentinel-3 satellite data with a constrained physical forward model

Hannah Niehaus[1], Larysa Istomina[1,2], Marcel Nicolaus[2], Ran Tao[2], Alexey Malinka[3], Eleonora Zege[3], and Gunnar Spreen[1]

[1]Institute of Environmental Physics, University of Bremen, Bremen, Germany
[2]Alfred Wegener Institute, Helmholtz Centre for Polar and Marine Research, Bremerhaven, Germany
[3]Institute of Physics, National Academy of Sciences of Belarus, Minsk, Belarus

**Correspondence:** Hannah Niehaus (niehaus@uni-bremen.de)

**Abstract.** The presence of melt ponds on Arctic summer sea ice significantly alters its albedo and thereby the surface energy budget and mass balance. Large-scale observations of melt pond coverage and sea ice albedo are crucial to investigate the role of sea ice for Arctic amplification and its representation in global climate models. We present the new Melt Pond Detection 2 (MPD2) algorithm, which retrieves melt pond, sea ice, and open ocean fractions as well as surface albedo from Sentinel-3 visible and near-infrared reflectances. In contrast to most other algorithms, our method uses neither fixed values for the spectral albedo of the surface constituents nor an artificial neural network. Instead, it aims for a fully physical representation of the reflective properties of the surface constituents based on their optical characteristics. The state vector $X$, containing the optical properties of melt ponds and sea ice along with the area fractions of melt ponds and open ocean, is optimized in an iterative procedure to match the measured reflectances and describe the surface state. A major problem in unmixing a compound pixel is that a mixture of half open water and half bright ice cannot be distinguished from a homogeneous pixel of darker ice. In order to overcome this, we suggest to constrain the retrieval with a priori information. Initial values and constraint of the surface fractions are derived with an empirical retrieval which uses the same spectral reflectances as implemented in the physical retrieval.

The snow grain size and optical thickness are changing with time and thus the ice surface albedo changes throughout the season. Therefore, field observations of spectral albedo are used to develop a parameterization of the sea ice optical properties as a function of the temperature history of the sea ice. With this a priori data, the iterative optimization is initialized and constrained, resulting in a retrieval uncertainty of below 8 % for melt pond and 9 % for open ocean fractions compared to the reference dataset. As reference data for evaluation, a 10 m resolution product of melt pond and open ocean fraction from Sentinel-2 optical imagery is used.

## 1 Introduction

In the central Arctic, summer starts , depending on latitude, in May and lasts until late August or early September. This period is defined by the presence of solar shortwave radiation rapidly warming the air and sea ice surface. The heat uptake by the ocean is limited by the sea ice cover which acts as a protective shield against the incoming solar radiation because of its high



albedo. Thick, white ice or even snow covered ice reflects more than 70 % of the incoming radiation back into the atmosphere
(Malinka et al., 2018; Light et al., 2022), while the darker, open ocean would absorb 90 % of the solar energy if it was not
protected by sea ice (Pohl et al., 2020). Thus, changes in sea ice extent, thickness or albedo have a strong impact on the Arctic
energy budget (Fetterer and Untersteiner, 1998; Perovich et al., 2002; Nicolaus et al., 2012). Melt ponds, which form on the
Arctic sea ice surface once the melting point is exceeded, drastically lower the surface albedo (Eicken et al., 2004; Istomina
et al., 2015a; Light et al., 2022) and lead to an increased absorption of solar radiation in the sea ice and upper ocean (Perovich
et al., 2003). This in turn accelerated the sea ice melt and is known as the sea ice-albedo feedback mechanism (Curry et al.,
1995; Perovich et al., 2008; Wendisch et al., 2023). Due to this feedback mechanism, some studies find that melt ponds could
be used to predict the seasonal Arctic sea ice extent minimum in September (Schröder et al., 2014; Liu et al., 2015), which
has been dramatically declining in the past decades (Stroeve et al., 2012b, a; Screen, 2021). Additionally, melt ponds strongly
impact the Arctic ecosystem, because the increased amount of available photosynthetically active radiation in and beneath the
ice enhances primary production (Frey et al., 2011; Light et al., 2015; Katlein et al., 2019; Nicolaus et al., 2022).

Field observations help understanding the formation processes and seasonal evolution of melt ponds (Yackel et al., 2000;
Perovich et al., 2002; Eicken et al., 2004; Polashenski et al., 2012; Webster et al., 2022). However, these studies also show high
temporal and spatial variability of melt pond formation and coverage, especially on undeformed first year ice (Scharien and
Yackel, 2005). This variability is challenging but necessary to be represented in global climate models (Flocco and Feltham,
2007; Dorn et al., 2018; Hunke et al., 2013; Zhang et al., 2018) to realistically simulate the albedo and mass balance of Arc-
tic sea ice. Therefore it is essential to analyze the large-scale distribution and evolution of melt ponds and thereby quantify
the importance of sea ice in terms of Arctic Amplification and its impact on the global climate system (Serreze et al., 2009;
Wendisch et al., 2023). For this purpose, satellite observations are the only suitable measurements. They can cover the entire
Arctic on a regular basis, at coarser resolutions of 500 m to 1.2 km even daily. The first pan-Arctic melt pond fraction product
was developed by Tschudi et al. (2008) using MODIS (Moderate Resolution Imaging Spectroradiometer) spectral surfaces
reflectances in the optical range. They use fixed reflectance values for the surface type components though, which does not
account for the high variability of sea ice and melt pond optical properties, which can lead to misclassification and larger un-
certainties. Following approaches generating pan-Arctic melt pond fraction datasets mostly involve Artificial Neural Networks
(Rösel et al., 2012; Ding et al., 2020; Lee et al., 2020; Feng et al., 2022; Peng et al., 2022). However, there are discrepancies
of more than 10 % between melt pond fraction products (Lee et al., 2020) and the artificial character of the retrievals impedes
the analysis and understanding of the physical reasons. Zege et al. (2015) have developed a physical retrieval, which is based
on field observations of melt pond and sea ice spectra (Polashenski et al., 2012; Istomina et al., 2013) and the representation of
the bi-directional reflectance of the surface as a function of its physical properties as suggested in Malinka et al. (2016, 2018).
This approach enables a better assessment of retrieval performance (Istomina et al., 2015a, b) and adjustments with regard to a
changing Arctic without the need of new training data.

We present a new algorithm, called MPD2, that builds on the implementation of the physical MPD1 retrieval by Zege et al.
(2015); Istomina et al. (2015a, b). While MPD1 was developed for the distinction between two surface types, sea ice and melt
ponds, only, we add a third, open ocean, surface type class to avoid a systematic overestimation of melt ponds. Knowledge



gained from field observations of summer sea ice and melt ponds is used to initialize and constrain the retrieval and thereby

afford the additional free parameter. A parameterization is developed from *in-situ* spectral albedo measurements to estimate sea ice properties as a function of the air temperatures the sea ice has undergone. This is motivated by the temporal sea ice albedo change which also depends on the temperature (changes in snow grain size and optical thickness). As we are retrieving sub-satellite-pixel surface fractions, such unaccounted albedo changes of the sea ice/snow surface type would negatively impact the retrieval uncertainty. Additionally, we have developed an empirical retrieval of the three surface types, which serves as

an initial step to provide constrained surface fraction values as starting guess for the physical melt pond retrieval. This initial empirical retrieval is based on the difference between the spectral signals of the surface types (Tschudi et al., 2008; Rösel et al., 2012; Istomina et al., 2015a; Light et al., 2022). The MPD2 algorithm is implemented for the Sentinel-3 satellite sensors OLCI (Ocean and Land Color Instrument) and SLSTR (Sea and Land Surface Temperature Radiometer) using top of the atmosphere (TOA) measured reflectances in the visible and near-infrared range at a spatial resolution of 1.2 km. Comparisons are drawn to

the MPD1 algorithm (Istomina et al., 2023) and to a 10 m resolution melt pond and open ocean fraction product derived from Sentinel-2 optical data (Niehaus et al., 2023). The focus of this study about the new MPD2 retrieval is the melt pond fraction product, which is also mainly evaluated here. However, along with melt pond fraction, the open ocean fraction and surface albedo are retrieved.

## 2  Datasets

The central algorithm presented in this work is based on optical data acquired by the Sentinel-3 satellites. For the processing, in addition, the low resolution sea ice drift product of the EUMETSAT Ocean and Sea Ice Satellite Application Facility (OSI SAF, www.osi-saf.org) and the reanalysis 2-m air temperature product by ERA5 are used. For the development, *in-situ* spectral albedo measurements are investigated. The evaluation of the presented algorithm is performed based on the melt pond fraction product derived from optical Sentinel-2 satellite data.

### 2.1  Sentinel-3 top of the atmosphere reflectances

The retrieval algorithm uses top of the atmosphere (TOA) measurements supplied by the EU Copernicus Sentinel-3A and -3B satellites operated by the European Space Agency (ESA) and European Organisation for the Exploitation of Meteorological Satellites (EUMETSAT). The two satellites orbit the Earth, phased by $180\,^\circ$, at an altitude of approximately 815 km in a polar, sun-synchronous orbit since February 2016 and April 2018, respectively. With a swath width of 1270 km and a short revisit

time, they cover the full Arctic, north of $67\,^\circ$, every day. However, optical observations are compromised by prevalent cloud contamination typical for the Arctic summer. Each of the two satellites carries multiple instruments, two of which are used in this work: The Ocean and Land Color Instrument (OLCI) and the Sea and Land Surface Temperature Radiometer (SLSTR) instrument. The OLCI instrument measures the Earth reflectance in 21 spectral bands in the visible and near-infrared (NIR) range (400 nm-1020 nm) with a spatial resolution of 1.2 km in reduced resolution mode and 300 m in full resolution mode. In

this study we use the reduced resolution data in favor of computing time and because the complications of sub-pixel surface



type features apply to both resolutions. The SLSTR instrument measures TOA radiances and brightness temperatures in 9 spectral bands in a range from 554 nm to 12 $\mu$m with a spatial resolution of 500 m-1 km. In the Level-1 products, the data is provided as geolocated measurements for both instruments, for OLCI, additionally, the solar and observation angles are provided.

## 2.2 OSI SAF drift product

For the tracking of the sea ice motion, the OSI-405 low-resolution sea-ice drift product by EUMETSAT OSI-SAF (Ocean and Facility) is used. It provides sea ice motion vector fields for a nominal time span of 48 h merging single-sensor drift vectors derived from various satellite sensors with an optimal interpolation scheme. The product is a near real time product that is available daily since 2019 and with a grid spacing of 62.5 km, projected to the NSIDC polar-stereographic grid.

## 2.3 ERA5 temperature

Along the drift track of ice parcels determined with the OSI SAF drift data, we use the 2-m air temperature provided by the ECMWF fifth generation reanalysis ERA5 (Hersbach et al., 2020) to calculate a temperature index as described in section 4.2. The data is downloaded from the ERA5 webpage with a spatial resolution of 0.25 $^\circ$ and a temporal resolution of 6 h.

## 2.4 In-situ albedo observations of radiation stations

*in-situ* observations of spectralalbedo from various measurement campaigns (TARA campaign 2007 (Nicolaus and Gerland, 2022), PS106-ARK31/1 expedition 2017, AlertMAPLI18 campaign 2018 and MOSAiC campaign 2019-2020 (Nicolaus et al., 2022)) are used in this work to investigate the relation between surface properties and the air temperatures. The datasets were obtained by the use of autonomous platforms installed on drifting sea-ice. These platforms comprised two RAMSES spectral radiometers with a spectral resolution interpolated to 1 nm, covering wavelengths from 320 nm to 950 nm. They were mounted 1 m above the sea-ice surface, measuring the incoming solar irradiance and upward reflected solar irradiance. From these measurements spectral albedo is obtained and provided. More details on the measurement setup and instruments can be found in Nicolaus et al. (2010).

## 2.5 Sentinel-2 melt pond fraction

For the development and evaluation of the MPD2 algorithm, 10 m spatial resolution melt pond fraction data derived from Sentinel-2 satellite Level-1C optical imagery (Niehaus et al., 2023) is used. The dataset provides melt pond fraction and open ocean maps up to latitudes of 82.3 $^\circ$, covering areas of 1000 km$^2$ to 10000 km$^2$ each (Niehaus and Spreen, 2022). The provided Sentinel-2 scenes are filtered by the availability of reasonable, cloud-free Sentinel-3 data within a time difference of less than 1 h, which results in 33 remaining scenes. To enable a quantitative comparison, the data from Sentinel-2 is drift corrected to match the Sentinel-3 product by using the OSI-SAF drift product described in section 2.2.



## 3 The Melt Pond Detection 2 algorithm

The Melt Pond Detection 2 (MPD2) algorithm builds on the Melt Pond Detection 1 (MPD1) algorithm which was developed by Zege et al. (2015) to retrieve pan-Arctic melt pond fraction and surface albedo from optical satellite data. Initially, the algorithm was designed for the application to data from the MERIS (MEdium Resolution Imaging Spectrometer) sensor on ENVISAT (Environmental Satellite) which was operating from 2022 to 2012. The adaption for the application to the data acquired by the Sentinel-3 satellites is described in Istomina et al. (2023). In contrast to most other pan-Arctic retrievals, e.g., by Rösel et al. (2012); Ding et al. (2020); Lee et al. (2020); Feng et al. (2022); Peng et al. (2022), the MPD algorithm is not based on the implementation of Artificial Neural Networks but uses a radiative transfer model representing the reflective properties of melt ponds and white ice based on their physical characteristics. Herein, the white ice category comprises various types of the sea ice surface combined by their high albedo and white appearance. This includes bare sea ice with a thin but highly scattering surface layer ontop, which is formed after meltwater has drained (Tschudi et al., 2008; Malinka et al., 2016), as well as snow in its different melting stages. These surfaces differ by their grain size, optical thickness and other properties which, however, are all accounted for in the same metrics by Zege et al. (2015), so that sea ice and white ice are used synonymously in the following. The bi-directional reflectance distribution function (BRDF) (of Standards and Nicodemus, 1977) of the snow/white ice/melt pond surface is used instead of the albedo, to account for the special illumination conditions of low solar elevation angles in the Arctic as well as of different observation angles. This is important as the occurring sun glint significantly contributes to the surface albedo but is not captured by the satellite observing at high elevation angles. The total BRDF $R$ of a pixel at the sea level is a linear combination of the surface components $R_{wi}$, $R_{mp}$ and $R_{oc}$, weighted by their surface fractions $s_{wi}$, $s_{mp}$ and $s_{oc}$, for white ice, melt pond and open ocean, respectively:

$$R = s_{wi} \cdot R_{wi} + s_{mp} \cdot R_{mp} + s_{oc} \cdot R_{oc}. \tag{1}$$

The TOA relfectance is then calculated by modeling radiative transfer through the atmosphere. The novelty of the MPD2 algorithm is the representation of open ocean as a third surface type class, along with the sea ice and melt pond classes that were considered in the MPD1 algorithm already. The area fractions of these three surfaces are related by the following equations:

$$s_{wi} + s_{mp} + s_{oc} = 1 \tag{2}$$
$$s_{wi} = (1 - f_{mp}) \cdot s_{ic} \tag{3}$$
$$s_{mp} = f_{mp} \cdot s_{ic} \tag{4}$$

Herein, it is important to distinguish between the melt pond surface area fraction $s_{mp}$ that is given with respect to the total area of a pixel and the melt pond fraction $f_{mp}$ which is considered with respect to the total area of sea ice $s_{ic}$. The total area of sea ice $s_{ic}$ combines the area fraction of melt ponds $s_{mp}$ and the area fraction of white ice $s_{wi}$, which is the ice area that is not covered by melt ponds.

Because of the close to specular reflectance of the open ocean which is, as mentioned above, not captured by the satellite due





the low sun elevation and high viewing angle of the satellite, the radiance reflected by the open ocean is neglected for the BRDF calculation. With this assumption and the Equations (2), (3) and (4), Equation (1) can be transformed to:

$$R = (1 - s_{oc}) \cdot f_{mp} \cdot R_{mp} + (1 - s_{oc}) \cdot (1 - f_{mp}) \cdot R_{wi}. \tag{5}$$

The area fractions $s_{oc}$ and $f_{mp}$ are part of the retrieval output.

The BRDF of the white ice $R_{wi}$ is calculated from the inherent optical properties of a stochastic medium using the model of random mixture by Malinka (2014), resulting in three dominant physical parameters: The optical thickness $\tau_{wi}$ of a surface layer, the mean effective grain size $a_{eff}$ and the absorption coefficient of yellow matter $\alpha_y$ deposited on the ice surface. The optical thickness of the sea ice describes the ability of light penetrating into the ice without being absorbed and is defined by

the sea ice texture. The effective grain size is defined as the mean chord length of ice in snow and white ice and is related to the widely used specific surface area (SSA) as

$$a_{eff} = \frac{4}{\rho_{ice} \cdot SSA}, \tag{6}$$

where $\rho_{ice}$ is the ice density.

To derive the BRDF of melt ponds $R_{mp}$, only isotropic reflection from the pond bottom is taken into account, neglecting

sideward reflections and deviations from the Lambertian law due to multiple reflections between pond bottom and surface. Additionally, the melt water is assumed to be clear, without any contaminants. By this, three further parameters of importance are determined: The pond depth $h_{pond}$, the ice thickness underneath the pond $h_{ice}$ and the transport scattering coefficient of that ice $\sigma_{ice}$, defining the spectral albedo of the pond bottom (Malinka et al., 2018).

In combination with the area fractions $s_{oc}$ and $f_{mp}$, these physical parameters dominating the ice and pond surface reflections,

constitute the state vector

$$\boldsymbol{X} = (f_{mp}, s_{oc}, \tau_{wi}, a_{eff}, \alpha_y, h_{pond}, h_{ice}, \sigma_{ice}), \tag{7}$$

which describes the state of the surface and is optimized in an iterative procedure to find the state that models the measured TOA reflectances the best. In Table 1 an overview of the parameters and their dimensions is given.

**3.1 The algorithm structure**

The input data to the algorithm is acquired by the Sentinel-3A and -3B satellite instruments OLCI and SLSTR introduced in section 2.1 at a spatial resolution of 1.2 km. From the OLCI instrument the TOA radiances $R_i$ of those channels (Table 2) are used that correspond to the wavelengths of the channels used previously by the MERIS sensor for MPD1 **?** Additionally, the viewing and illumination zenith and azimuth angles are entered into the algorithm. For cloud screening, the channels S7, S8

and S9, with the wavelengths given in Table 2, from the SLSTR instrument are used (Istomina et al., 2023).

As auxiliary input data to set the initial values the allowed range of parameters in $\boldsymbol{X}$, the OSI-SAF drift product and the ERA5 2-m air temperature are used. This data is necessary to calculate the temperature history of every pixel and is inserted





**Table 1.** State vector $X$ of optimized parameters

| Symbol | Characteristics | Dimension |
|---|---|---|
| $f_{mp}$ | melt pond fraction with respect to ice area | dimensionless |
| $s_{oc}$ | open ocean fraction with respect to pixel area | dimensionless |
| $\tau_{wi}$ | optical thickness of white ice | dimensionless |
| $a_{eff}$ | effective grain size of ice surface | $\mu$m |
| $\alpha_y$ | absorption coefficient of yellow matter | m$^{-1}$ |
| $h_{pond}$ | depth of melt pond | m |
| $h_{ice}$ | ice thickness beneath melt pond | m |
| $\sigma_{ice}$ | transport scattering coefficient of pond bottom | m$^{-1}$ |

**Table 2.** OLCI and SLSTR spectral channels used. Channels near 550 nm are excluded in order to avoid the effect of ozone absorption.

| | # | center wavelength [nm] |
|---|---|---|
| | 2 | 412.5 |
| | 3 | 442.5 |
| | 4 | 490 |
| | 10 | 681.25 |
| OLCI | 12 | 753.75 |
| | 16 | 778.75 |
| | 17 | 865 |
| | 18 | 885 |
| | S7 | 3742 |
| SLSTR | S8 | 10854 |
| | S9 | 12022.5 |

into the process block where the initial values and boundaries for the state vector $X$ are set. The details of this procedure are the core of this study and will be discussed in detail in the Sections 4 and 5.

To reduce the input data to pixels where the algorithm is applicable, first a landmask is applied and secondly pixels with cloud contamination are filtered out. To achieve this, cloud screening criteria introduced by Istomina et al. (2010, 2011) and the differential snow index criterion (Zege et al., 2015) are applied. Atmospheric corrections are applied by the use of the radiative transfer code RAY (Tynes, 2001) to take into account scattering and absorption by aerosols and gases, as well as molecular scattering and is explained in detail in (Zege et al., 2015). The atmospheric correction block calculates for every

pixel the transmission $t_i$ and the reflection $r_i$ of the atmosphere in the spectral bands $i$ of the used wavelengths. With these, the maximal possible TOA reflectance $R_{max}^{TOA}$, in case of perfect reflection at the surface, is calculated. This value is used in



the iterative procedure as a threshold. These values, $t_i$, $r_i$ and $R_{max}^{TOA}$, are passed to the iterative procedure in combination with the initialization and the boundaries of the state vector $\boldsymbol{X}$ for all the pixels that are determined to be reasonable pixels for the algorithm. The following description of the iterative procedure is visualized in the flowchart in Figure 1. It is explained for a single pixel as the processing is independent between pixels.

1. Starting with the initialized state vector $\boldsymbol{X}$ describing the surface, the TOA reflectances that would be expected are calculated

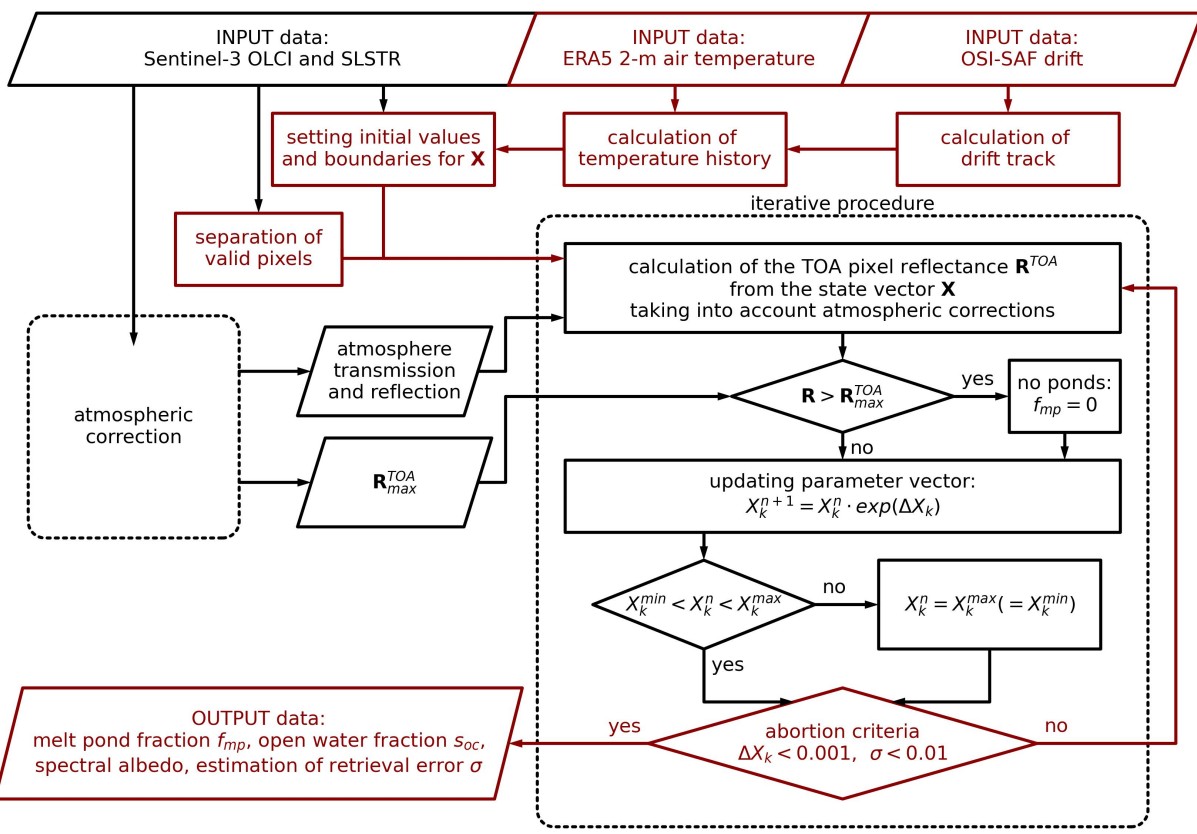

**Figure 1.** Flowchart of the MPD2 algorithm. The atmospheric correction block is left blank in favor of simplicity, the details can be found in Zege et al. (2015). The red color marks parts of the algorithm where major changes have been introduced compared to the MPD1 version. The current iteration step of the optimization process is denoted as $n$ and $k$ indicates the entries of the state vector $\boldsymbol{X}$.

using Equation 5, the forward model (Malinka, 2014; Malinka et al., 2018) for the BRDF of the white ice and melt ponds, and the transmission and reflectance coefficients of the atmosphere.

2. The measured TOA reflectances in the eight spectral channels $i$ are compared to the reflectance threshold $R_{max}^{TOA}$ from the atmosphric correction block for the respective channels. If the measured reflectance is higher than this threshold, $f_{mp}$ and $s_{oc}$ are zero because melt ponds or open ocean would reduce the measured TOA reflectance. Following, only the parameters



describing the white ice reflectance, $h_{pond}$, $h_{ice}$ and $\sigma_{ice}$, are included in further optimization steps. If the measured reflectance is below the threshold, all parameters of $\boldsymbol{X}$ are included.

3. The state vector is updated according to the differences between the modeled and measured TOA reflectances. This procedure

relies on the Newton-Raphson method (Press, 2007). For every component $X_k$ of the state vector and every channel $i$, the partial derivative of $R_i^{TOA}$ is calculated numerically to compose the matrix $\boldsymbol{M}$:

$$\boldsymbol{M} = (M_{ik}) = \left( X_k \cdot \frac{\delta R_i^{TOA}}{\delta X_k} \right). \tag{8}$$

With this matrix, the change applied to the components of $\boldsymbol{X}$ is determined from the current iteration step $n$:

$$\Delta X_k^{n+1} = X_k^n \cdot \mathrm{e}^{\Delta X_k}, \tag{9}$$

where the vector $\boldsymbol{\Delta X}$ is the product of the Moore-Penrose pseudoinverse (pinv) of matrix $\boldsymbol{M}$ (Press, 2007) and the difference vector between the measured TOA reflectances $\boldsymbol{R}$ and the TOA reflectances calculated in this iteration step $\boldsymbol{R^n}$:

$$\boldsymbol{\Delta X} = pinv\left(\boldsymbol{M}, \boldsymbol{\lambda_{min}}\right) \times \left(\boldsymbol{R} - \boldsymbol{R^n}\right). \tag{10}$$

4. If any updated value of $\boldsymbol{X}$ is beyond the respective boundaries, which were initialized before the iteration process, it is set to the passed boundaries and is not longer included in the remaining optimization process.

5. The steps 1.-4. are repeated until two abortion criteria are satisfied: The first requirement involves the intended change to the components of $\boldsymbol{X}$. If the condition

$$\Delta X_k < 0.001, \tag{11}$$

is met for all parameter $k$, it can be assumed that the algorithm has found a minimum where further iterations would not lead to significant changes of the output. The second condition takes the root mean squared difference $\sigma$ between the measured and

retrieved TOA reflectances into account to ensure that the minimum found can physically describe the surface state:

$$\sigma = \sqrt{\frac{1}{m} \sum_{i=1}^m (R_i - R_i^n)^2} < 0.01 \tag{12}$$

Finally, the output of the retrieval algorithm are the melt pond fraction, open ocean fraction and the spectral albedos at wavelengths 400 nm to 900 nm, in 100 nm steps. Additionally, $\sigma$ which is an estimator of the retrieval uncertainty for the specific pixel is provided.

## 3.2 Algorithm sensitivity to initial values of optimized parameters

The MPD2 algorithm is a complex, multiparameter retrieval. By including open ocean as third surface type class, the number of free parameters is increased to 8 (Table 1). In addition, open ocean is spectrally almost neutral with very little reflectance in the broadband optical and NIR spectrum (Pohl et al., 2020). This can result in a situation where a mixed pixel consisting of open ocean and bright ice is indistinguishable from a homogeneous pixel of darker ice and makes it virtually impossible to unmix



the spectral signals of the surface type components without additional information (Istomina et al., 2023). These additional information could be inserted into the algorithm by finding a suitable initialization and constraint of the state vector $X$. As the cost function minimized in the retrieval is expected to be multimodal, a suitable initialization of $\boldsymbol{X}$ offers potential to improve on the retrieval performance (as the optimization could otherwise run into a false local minimum in the distribution).

To investigate the algorithm sensitivity to the initial values of the optimized state vector $X$, a set of test pixels is selected. For

this selection the melt pond fraction product from Sentinel-2, described in Section 2.5 is used. After drift correction, the two satellite products are overlapping and the 10 m resolution Sentinel-2 product (where 120x120 pixel correspond to one of the Sentinel-3 pixels) is used to identify 6 cases of different dominating surface type classes and their combinations. For each of these 6 cases, 4 adjacent pixels of Sentinel-3 (corresponding to an area of 2.4 km x 2.4 km) are chosen yielding a testbed of 24 intentionally chosen pixels in total. Additionally, 6 cases of 4 pixels are chosen randomly. In Figure 2 the Sentinel-2 reference

melt pond fractions are displayed for all the 48 test pixels. In case of the intentionally chosen pixels, the present surface types in these pixels are given, ordered from highest to lowest fraction. For further discussion, we focus on these manually chosen pixels and average the results for the four pixels representing one case.

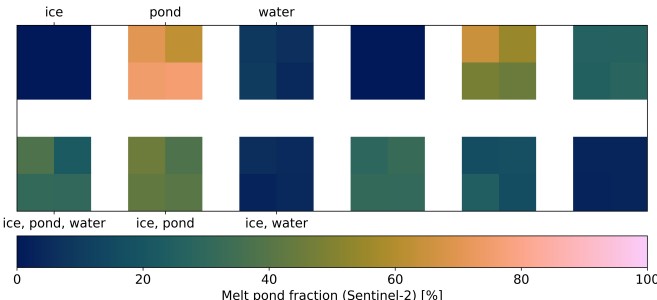

**Figure 2.** Sentinel-2 melt pond fraction reference of the test pixels. Half (right) of the pixels is chosen randomly, the other half (left) chosen with purpose to represent the typical surface types and their combinations as denoted by the labels. In this case, the present surface types in the pixels are listed ordered by their occurrence.

On the Sentinel-3 input data of these test pixels we run a Monte-Carlo simulation of the MPD2 algorithm to investigate the retrieval sensitivity to the initial values of the parameters in the state vector $\boldsymbol{X}$. All parameters $X_k$ are varied simultaneously

within according physical ranges. These ranges come from *in-situ* data and can be seen from the limits of the x-axes in Figure 3. This figure shows for each optimized parameter the probability density of retrieving a *reasonable* result as a function of the initial guess for the different surface type combinations. The definition of a *reasonable* result is

$$\frac{f_{mp}^{MPD2} - f_{mp}^{S2}}{1 + 2f_{mp}^{S2}} < 0.1, \qquad (13)$$

where $mpf_{MPD2}$ is the melt pond fraction retrieved with MPD2 and $mpf_{S2}$ is the reference melt pond fraction of the respective

pixel from Sentinel-2. Equation 13 combines the estimation of the absolute and relative error, so that for small values of $f_{mp}$



the absolute error needs to be less than 0.1, while for a 100 % melt pond surface the relative error should be below 0.3. Thus, this estimation smoothly transits from the relative difference to the absolute one at small values of $f_{mp}$.

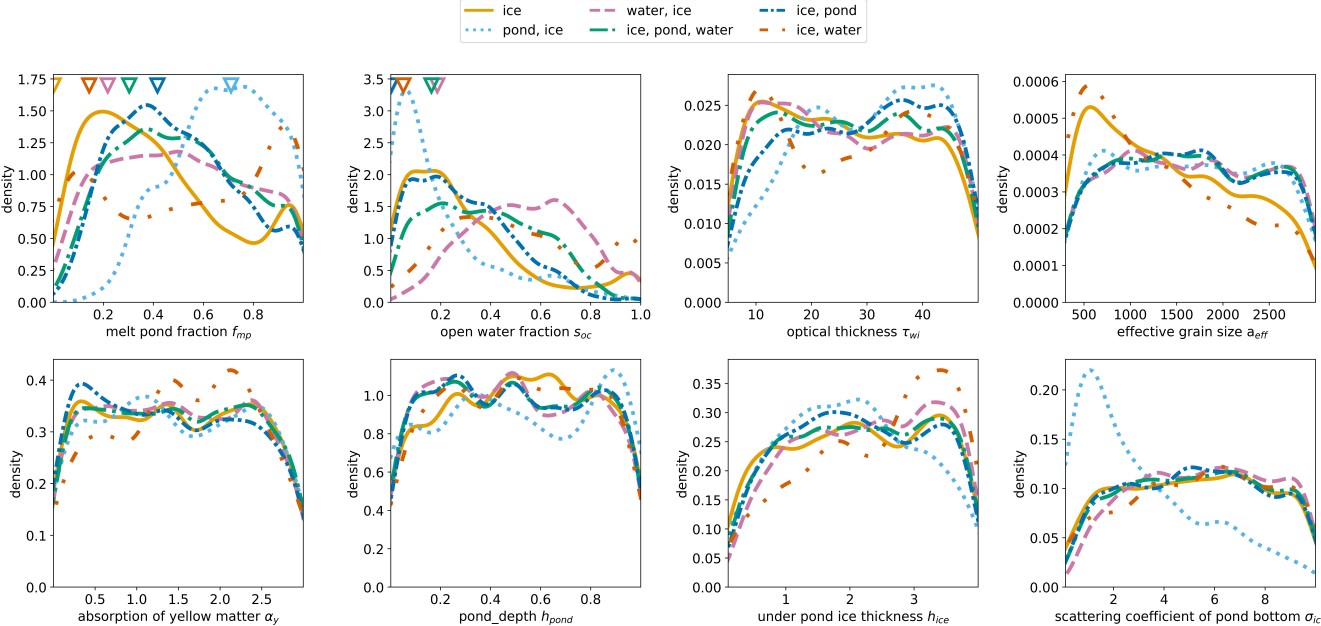

**Figure 3.** Probability densities of initial values yielding a *reasonable* result of melt pond fraction output. The reasonable output is defined by two criteria: (1) the difference between the melt pond fraction output and the reference value from Sentinel-2 is below 0.1 or 10 % and (2) the root mean square deviation (RMSD) of the retrieved reflectances compared to the measured reflectances is below 0.1. The eight panels show the distributions for the different optimized parameters of the state vector $X$. The differently colored lines represent the average of the four test pixels per dominant surface type (combinations). The triangles in the two upper left panels show the reference values of melt pond and open ocean fraction from Sentinel-2.

To analyze Figure 3 it is important to note that the probability densities are standardized to the number of *reasonable* results. There are two different ways to look at the graphs: 1. Either the focus is the difference between the curves representing the surface types. This enables an interpretation of how important it is to choose the initial values for every surface separately. 2. Or the focus is the dependency of a single curve or multiple similar curves on the initial value. This shows how important the initial values for this case (parameter, surface type) is and can help to choose suitable ones. If there is a strong peak the suitable initialization is important. If the curve is close to a uniform distribution, it is not.

For $f_{mp}$ and $s_{oc}$ in the two upper left panels of Figure 3, the differences between the various surface types and combinations are the strongest. The triangles at the top of these two panels give the reference values from the Sentinel-2 product. The comparison of the reference values and the maxima of the probability density curves lead to the conclusion that the retrieval works better when the initial value is already somewhat close to the reality. Interestingly, an initial value of $f_{mp} = 0$ is hardly ever good for the performance of the retrieval. However, it is obvious that the initial value and boundary of $f_{mp}$ and $s_{oc}$ should not be





fixed but adapted for every pixel to have a higher likelihood to achieve the best result. The effective grain size $a_{eff}$ of sea ice
is related to its albedo (Figure 6) and shows in the upper right panel a strong impact on the retrieval results for the *ice* and
the *ice, water* surfaces, the latter of which is also strongly dominated by ice. For the other surface types $a_{eff}$ seems to have
little impact on the retrieval performance. Another noticeable feature is the strong peak at small scattering coefficients $\sigma_{ice}$
for ponded pixels in the lower right panel, while all other surface types show almost a uniform distribution. This is neglected
because we expect the influence of the initial value for $\sigma_{ice}$ to decrease with other parameters being constrained. For the other
optimized parameters, $\tau_{wi}$, $\alpha_y$, $h_{pond}$, $h_{ice}$, the dependency of the probability of a *reasonable* result on the initial values is not
as strong.

Based on this analysis, we conclude that an approach is needed to initialize and constrain $f_{mp}$ and $s_{oc}$ as well as $a_{eff}$. In the
following two chapters constraints for 1. $a_{eff}$ and $\tau_{wi}$, which are correlated, and 2. $f_{mp}$ and $s_{oc}$ are developed.

## 4 Initial values for effective grain size $a_{eff}$ and optical thickness $\tau_{wi}$

The reflective properties of sea ice are highly variable. In April and May, most of it is covered by snow featuring a very high
albedo. This is because dry snow consists of small grains, typically around 100-300 $\mu$m (Jäkel et al., 2021), leading to high
(almost infinite) optical thickness and strong scattering (Perovich, 1979). When the air temperature approaches the melting
point, the surface becomes wetter and snow grains start to bond (Marsh, 1987), a process called sintering (Blackford, 2007).
Thereby the snow grain size increases and the optical thickness of the surface layer becomes finite. Once the snow is melted
and drained into surface depressions to form melt ponds, white ice is left. The reflection of this ice is defined by a surface
scattering layer (Grenfell and Maykut, 1977) and can be described by the same metrics as snow (Zege et al., 2015; Malinka
et al., 2016). However, the grain sizes are much larger and the optical thickness is reduced.

It is eminent that the effective grain size $a_{eff}$ and optical thickness $\tau_{wi}$ of a sea ice surface layer (consisting of white ice or
snow) are correlated and define the albedo. Moreover, they depend on the wetness of the surface and thus on the air temperature.
The goal of this section is to find a simple relation between the temperature history that the sea ice experienced and the
physical properties $a_{eff}$ and $\tau_{wi}$. For that purpose, the satellite spectral measurements of the surface reflectance are not longer
considered a snapshot in time but as the result of the previous history of that surface. That means the drift trajectory of every
single satellite pixel is calculated from the time of measurement back to the beginning of the melt season using the OSI-SAF
drift product described in section 2.2. Along this drift track the 2-m air temperatures are collected from the ERA5 product
described in section 2.3. The empirical relation found between the temperature history along the drift track and the optical
properties is based on *in-situ* spectral albedo measurements introduced in section 2.4. The development and filtering of this
data is described in the following section 4.1. Afterwards, in section 4.2, the correlation between temperature history and
optical surface properties is explored and discussed.



### 4.1 In-situ data preparation

We use spectral albedo measurements from *in-situ* radiation station data introduced in Section 2.4 to develop the relationship between temperature history and surface optical properties. It is necessary to make sure that only albedo measurements of sea ice and not of melt ponds are used for this analysis because $a_{eff}$ and $\tau_{wi}$ are no appropriate properties to describe melt pond reflection. For this purpose, we use a conservative threshold applied to the ratio of the albedo at $500\,\mathrm{nm}$ and $900\,\mathrm{nm}$, which is sensitive to the water content of the surface (Nicolaus et al., 2010; Tao et al., 2023). If the ratio is >1.5, the data is removed

from the evaluation because there most likely is a melt pond in the observation area. Then the forward model used in the MPD2 algorithm itself and described in Malinka et al. (2016) is used to derive $a_{eff}$ and $\tau_{wi}$ from the *in-situ* spectral albedo measurements. The model output comes along with an uncertainty estimate which is used to filter out measurements where the model was not able to reproduce the observed spectra sufficiently and thus the derived values of $a_{eff}$ and $\tau_{wi}$ are not reliable. Additionally, we use an averaged slope of the albedo curve in combination with the albedo at $400\,\mathrm{nm}$ to eliminate observations

where $a_{eff}$ is inexplicable high, probably because of irregular sky conditions of an undetected melt pond for which the model is not valid. Subsequently, events of fresh snowfall after the melting had already started are removed from the *in-situ* dataset as these are not covered by the forward model. This is done by the application of a lower threshold of $300\mu m < a_{eff}$ to the effective grain size according to Jäkel et al. (2021). Figure 4 shows the time periods where observations are available after these steps of filtering.

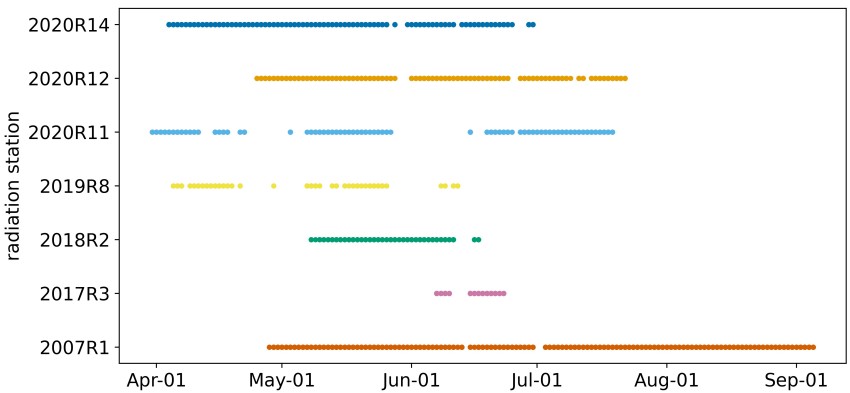

**Figure 4.** Overview of the available radiation station data throughout the summer period after data filtering. The first four digits of the station names indicate the year of observation.

### 4.2 Relation between the physical properties and the temperature history


To relate the modeled $a_{eff}$ and $\tau_{wi}$ to the temperature history dictating the current state of the sea ice surface we decided to use the approach of the Cumulative Melting Degree Day (CMDD) index, which is defined as the integrated temperature





above 0 °C over time. To be able to take into account short freezing periods in between the melting periods, we adapted the conventional CMDD index resulting in an index we call $T_{idx}$. Starting at the beginning of observations, or at the date where the backtracking of a pixel ends, the temperature is evaluated at every time step $t_i$ to add to or subtract from the current value of the temperature index $T_{idx}$. It is important that this starting date of calculations lies before the melt onset. Figure 5 shows the iterative procedure that is used to gain the temperature index $T_{idx}$ at any time of observation $t_{obs}$. This procedure takes a list of temperatures along the back projected drift trajectory of a pixel (observation) and comprises two blocks, one handling the melting temperature (left, red part), the other handling the freezing periods (right, blue part). If the temperature is above 0 °C the common definition of cumulative melting degree days (CMDD) is used:

$$CMDD = \int_t T \cdot dt. \tag{14}$$

When the temperature drops below 0 °C, $T_{idx}$ is reduced in proportion to the time period, magnitude of negative temperature and the current value of $T_{idx}$. This approach is based on the assumption of the surface getting drier when it is freezing again, stopping the sintering and leading to a slight refinement of the grains. As soon as the temperature rises again above 0 °C, we switch back to the increase of $T_{idx}$ following the CMDD definition in Equation 14.

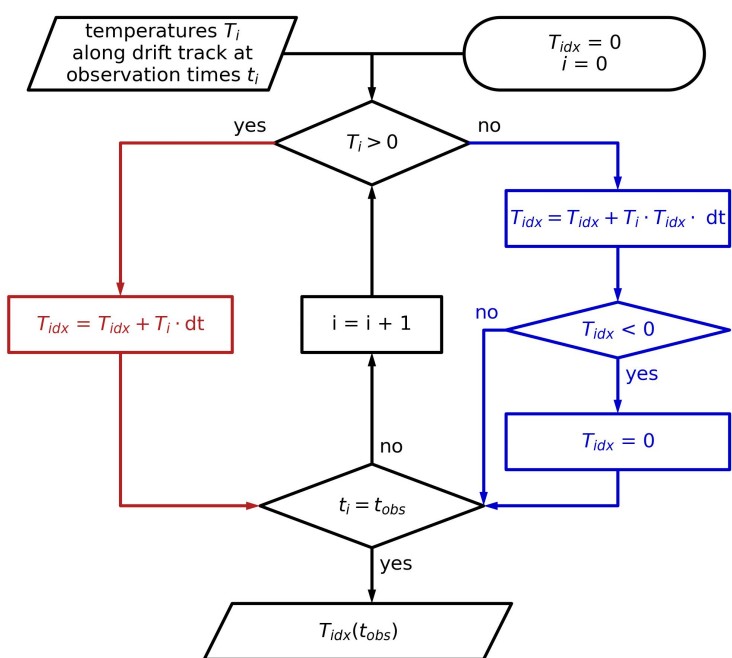

**Figure 5.** Flowchart of the calculation of the temperature index $T_{idx}$ at any time of observation $t_{obs}$. $i$ iterates through the times steps of size $dt$. As $T_{idx}$ is initialized at 0, it is important that the beginning of this iteration (observation) period lies before the start of any melting.





As an example, Figure 6 shows the modeled physical quantities $a_{eff}$ and $\tau_{wi}$ together with the 2-m air temperature $T$ and the derived temperature index $T_{idx}$ for the radiation station *2020R12*. With an overall increasing $T_{idx}$, the optical thickness of the surface layer is decreasing while the effective grain size is increasing. Strong fluctuations are especially visible for the modeled grain size corresponding to different amounts of scattering layer. There are two distinct data gaps in the modeled data

at the end of May and around June 25. This is where the data has been filtered out because of the likely presence of melt ponds (Nicolaus et al., 2010). Around these two gaps, the scatter is especially high showing the strong surface variability close to melting and freezing conditions.

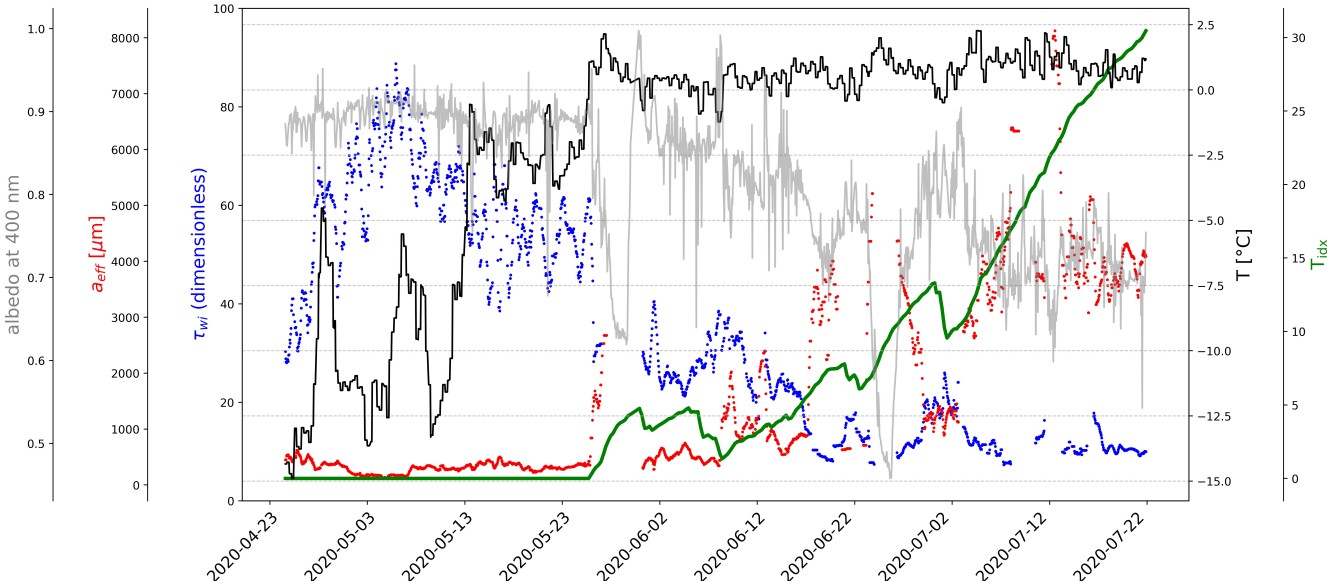

**Figure 6.** Effective grain size $a_{eff}$ (red) and optical thickness $\tau_{wi}$ (blue) retrieved from the spectral albedo measurements of the radiation station *2020R12*, 2-m air temperature (black) from ERA5, and the temperature index $T_{idx}$ (green) derived from the temperature.

The physical properties $a_{eff}$ and $\tau_{wi}$ as a function of the temperature history are shown in figures 7 and 8. Here, the observations of all measurement stations are combined, with the color indicating the radiation station according to the color

code in Figure 4. We use the following fit functions to describe their relations:

$$a_{eff} = a \cdot \log(b + T_{idx}) + c \tag{15}$$

$$\tau_{wi} = a \cdot \exp(-b \cdot T_{idx}) + c \tag{16}$$

These fits are applied to the full dataset to determine the initial value of the respective parameter as a function of the temperature index. This initial value function we will later use in the MPD2 retrieval. To get upper and lower boundaries also as a

function of $T_{idx}$, we use the 95 and 5 percentiles of the data. The resulting fit parameters a,b and c for both properties are given in Table 3.

In Figure 7, $a_{eff}$ features low values at $T_{idx} = 0$. With increasing $T_{idx}$ also $a_{eff}$ increases as well as the scatter, which is





covered by the increasingly wider range of allowed values marked by the light gray area. For $\tau_{wi}$ there is a lot of scatter visible at $T_{idx} = 0$ in Figure 8. This is covered by the wider range of allowed values marked by the light gray area. With increasing

$T_{idx}$, the variability gets less and the optical thickness converges at a lower value. Overall the dependence is not as strong as for $a_{eff}$ which is expected from the sensitivity analysis in section 3.2.

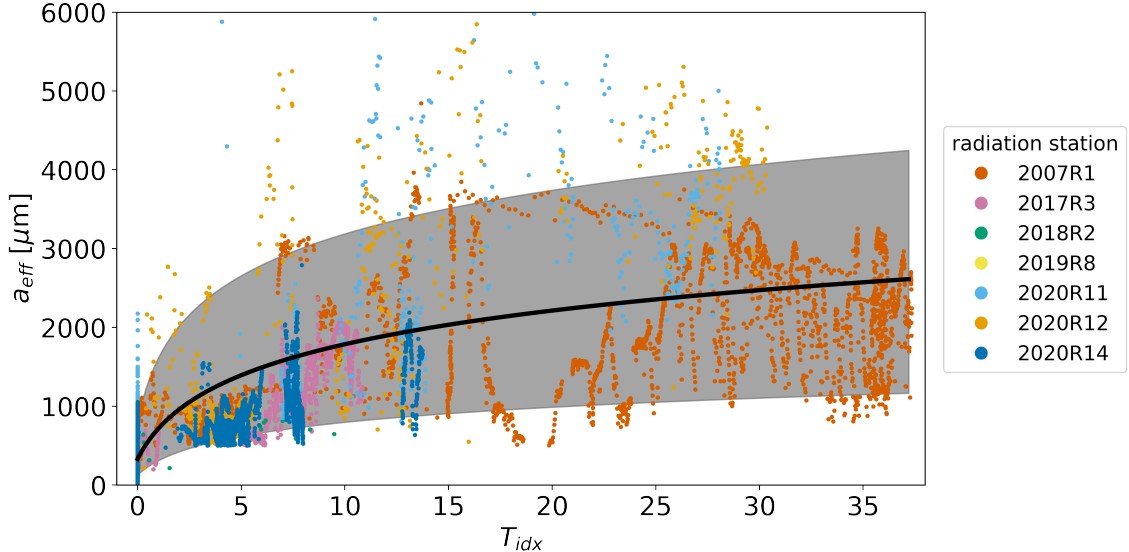

**Figure 7.** Scatter plot of modeled optical thickness $a_{eff}$ depending on the temperature index $T_{idx}$. The different colors indicate measurements of different radiation stations. The thick black line is the fit to all data, which we later use to determine the initial value of $a_{eff}$ depending on $T_{idx}$. The gray area marks the allowed values limited by the upper and lower boundaries derived from the 95 and 5 percentiles.

**Table 3.** Fit parameters to describe initial values and lower and upper boundaries for the optical thickness $\tau_{wi}$ and the effective grain size $a_{eff}$.

|  |  | Initial guess | Lower boundary | Upper boundary |
|---|---|---|---|---|
|  | a | 685.99 | 297.95 | 865.63 |
| $a_{eff}$ | b | 1.37 | 1.17 | 0.66 |
|  | c | 118.73 | 77.13 | 1142.93 |
|  | a | 23.18 | 8.69 | 49.27 |
| $\tau_{wi}$ | b | 0.27 | 0.23 | 0.36 |
|  | c | 12.24 | 8.20 | 16.35 |



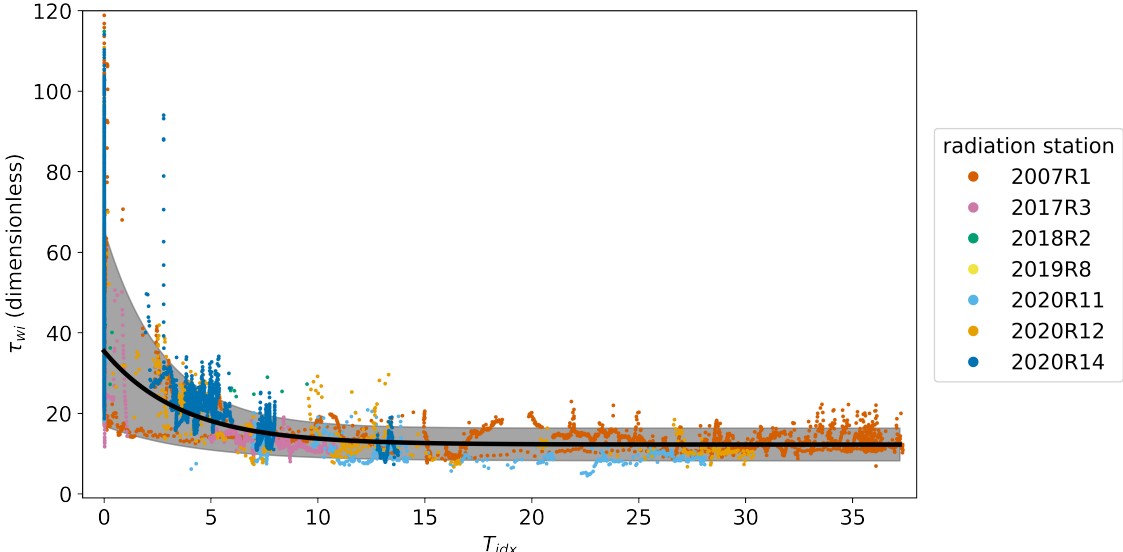

**Figure 8.** Scatter plot of modeled optical thickness $\tau_{wi}$ depending on the temperature index $T_{idx}$. The different colors indicate measurements of different radiation stations. The thick black line is the fit to all data, which we later use to determine the initial value of $\tau_{wi}$ depending on $T_{idx}$. The gray area marks the allowed values limited by the upper and lower boundaries derived from the 95 and 5 percentiles.

## 5 Initial values for melt pond $f_{mp}$ and open ocean $s_{oc}$ fraction

The spectral albedo of sea ice, melt ponds and open ocean features large variability. Even though the transition of the spectra
going from wet ice surfaces to light and shallow melt ponds is relatively fluent, there is great potential in the spectral measurements to differentiate between the difference surface types given they are observed separately. Whereas melt ponds have a much lower albedo in the NIR (0.1–0.2) than in visible wavelengths (0.4–0.7), unponded, dry ice shows little changes in albedo in the visible bluerange (0.7–1.0) and only a slight decrease towards the NIR (0.6–0.9) (Istomina et al., 2015a; Malinka et al., 2018; Light et al., 2022). As this spectral behavior is dictated by the amount of liquid water in the surface layer, the spectrum
of sea ice gets closer to the spectrum of a light melt pond, the wetter it is. Similar applies to wet snow which has a lower albedo in NIR and complicates distiguishing the surface types even more. Open ocean has a constant low albedo of below 0.1 (Pohl et al., 2020) in the visible and NIR range. However, at the 1.2 km spatial resolution of the Sentinel-3 sensors, as the situation becomes sub-pixel, it is complicated to disentangle the spectral signals of the mixed surface types (Istomina et al., 2023), which already themselves feature large variability. In particular, different compositions of surface fractions of ice, melt ponds, and
open ocean can result in the same mixed spectral signal at 1.2 km resolution if boundary conditions for, e.g., the snow grain size (larger grains reduce the albedo of the ice surface type), are not taken into account. Yet, there is the potential to get a *good first guess* of the composition of the surface types, which will be sufficient as initialization of the physical MPD2 algorithm. In this chapter, we develop a first order empirical retrieval of melt pond and open ocean fractions to use as initial values for the MPD2 algorithm. This empirical retrieval is based on the same TOA reflectances also fed into the physical MPD2 retrieval



algorithm. The empirical retrieval is developed on the 48 test pixels described in Section 3.2 and comprises two steps. First, we define the brightness $h$ of a pixel as the average TOA reflectance in the spectral bands shown in Figure 9. This value is used to estimate the total water fraction in the pixel without distinguishing between melt ponds and open ocean. This will be discussed in the following Section 5.1. Second, we define the slope of the spectrum from the reflectance difference between 490 nm and 754 nm. From this value the proportion of melt ponds and open ocean relative to the total water fraction is estimated, as presented in Section 5.2.

Note that in spite of the difference between albedo and reflectance, as well as between *in-situ* and TOA measurements, we neglect bothdifferences in this section because we only estimate the initial values and constraints empirically and the more accurate values will be achieved in the retrieval iteration cycle.

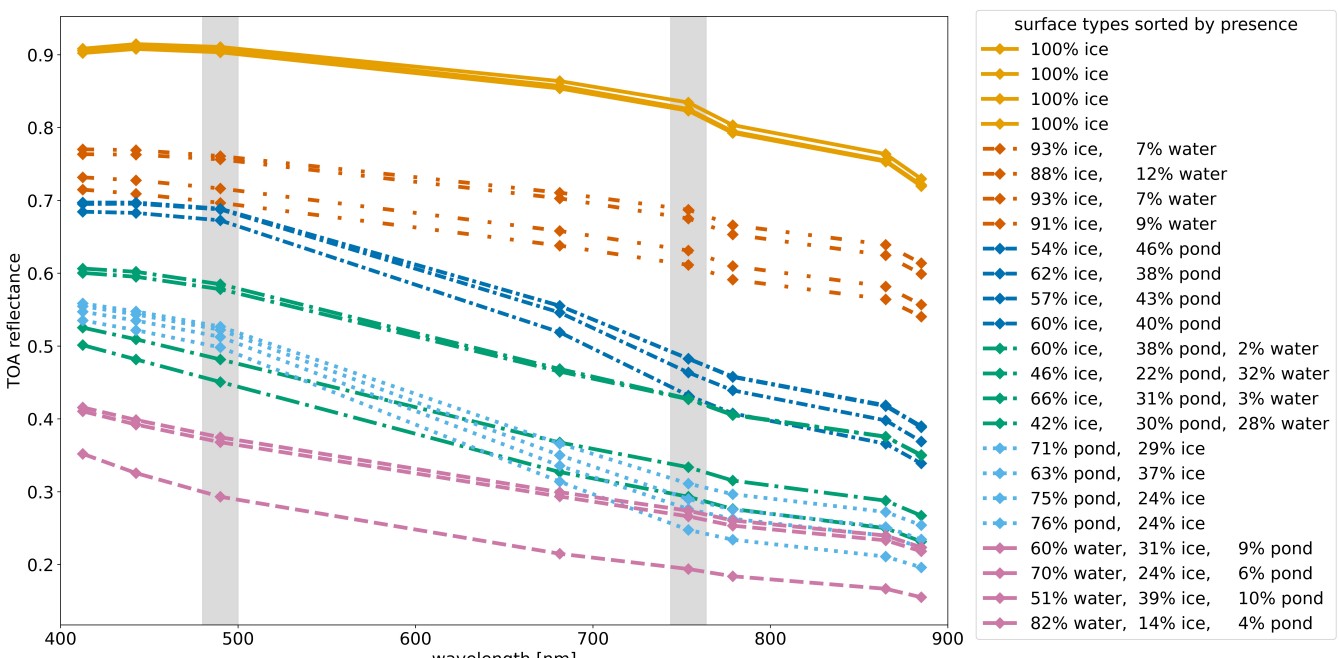

**Figure 9.** TOA reflectance spectra measured by the OLCI instrument on the Sentinel-3 satellite for the 24 test pixels with known dominant surface types (Figure 2, left part). These are indicated by the color and style of the lines connecting the measurements at eight different wavelengths. Additionally, the percentage of the surface types, known from the Sentinel-2 reference product, is given. The vertical gray areas mark the spectral bands used for the definition of the slope of the spectra in the empirical pre-retrieval.

### 5.1 Brightness criterion: Estimation of total water fraction

The brightness $h$ of a pixel is defined as the TOA average reflectance in the eight spectral bands shown in Figure 9, which includes atmospheric influence and serves for a rough empirical estimate: As sea ice is a much better reflecting surface than melt ponds or water, this quantity can be used to separate water and ice surfaces and thereby estimate the total water fraction



$twf$. This is the area fraction of a pixel that is covered either by melt ponds or by the open ocean. We set a fixed threshold $h_{min}$ on the brightness below which we assume the pixel to contain only water and no sea ice that is not covered by ponds.

The upper threshold $h_{max}$ above which the pixel contains sea ice only, is expected to depend on the time of the year and region of observation. This is because of the evolution of sea ice reflective properties with time and temperature (Marsh, 1987). Thus, we use again the temperature index $T_{idx}$ defined in Section 4.2, as the temperature describes the seasonal cycle and regional differences while observing exceptional weather situations. With increasing $T_{idx}$, the sea ice surface is comprising larger grain sizes i.e. becoming wetter and thus less reflecting but still to be considered as pure ice surface:

$$h_{min} = 0.175 \tag{17}$$

$$h_{max} = 0.75 - 0.002 \cdot T_{idx}. \tag{18}$$

The dependency in Equation 18 is determined from the *in-situ* spectral albedo measurements described in Sections 2.4 and 4.1. The brightness $h$ of the measurements is calculated from the wavelengths matching the spectral channels of the OLCI instrument used in the MPD2 algorithm. Then a linear fit is applied to extract a simple dependency and define a limit above

which it can be assumed that there is only sea ice observed. The inlay in figure 10 shows a density plot of the brightness $h$ as a function of $T_{idx}$ and the linear fit.

Between the lower $h_{min}$ and the upper threshold $h_{max}$, corresponding to a pure water and pure sea ice pixel, respectively, a linear transition is assumed:

$$twf = \frac{1}{h_{min} - h_{max}} \cdot h + \frac{h_{max}}{h_{max} - h_{min}}. \tag{19}$$

This yields the total water fraction $twf$ as a funciton of the pixel brightness $h$ as shown in Figure 10.

## 5.2 Slope criterion: Separation of melt ponds and open ocean

The slope $s$ for a satellite pixel is defined as the absolute ratio between the spectral reflectances at 490 nm and 754 nm. These two channels are marked by the gray vertical areas in Figure 9. The choice of these channels is based on the well investigated reflectance difference of sea ice and melt pond between the blue and red/NIR range (Rösel et al., 2012; Wang et al., 2020). We

decided to use the channel at 753.75 nm instead of any other channel because for longer wavelength the ice reflectance shows a stronger decrease, which would reduce the difference to the melt pond class (Figure 9). Based on $s$ we now derive the pond fraction $pf$ and the ocean fraction $of$. These quantities are defined relative to the total water fraction $twf$ and not to the pixel. Thus, the following explanation can be cut to the derivation of $pf$ only because $of$ is then defined by:

$$of = 1 - pf. \tag{20}$$

As shown in Figure 11, we assume a linear transition based on $s$, between the case of a 100 % ocean covered pixel (below $s_{min}$) and a 100 % melt pond covered pixel (above $s_{max}$):

$$pf = \frac{1}{s_{max} - s_{min}} \cdot s + \frac{s_{min}}{s_{min} - s_{max}}, \tag{21}$$



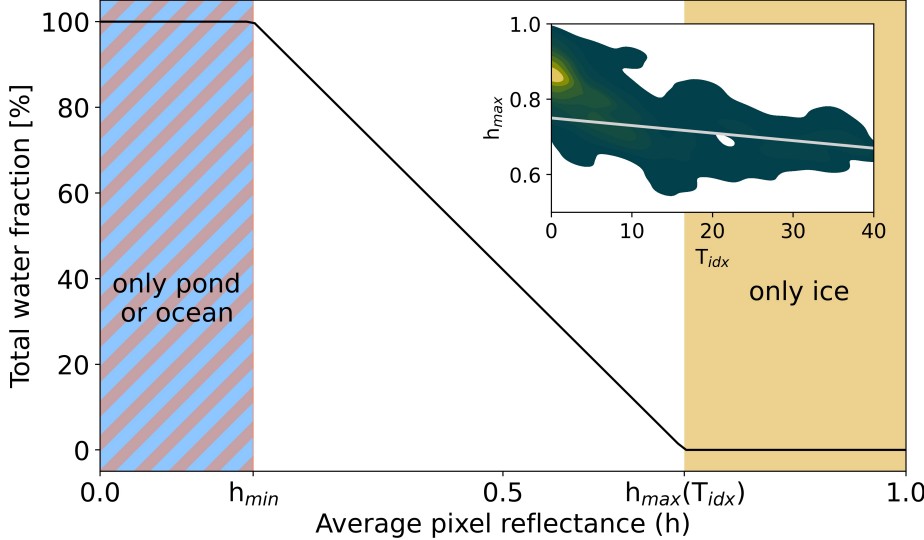

**Figure 10.** Main figure: total water fraction $twf$ as a linear function of the pixel brightness $h$ (average of used TOA reflectances) between the lower threshold $h_{min}$ below which the pixel is fully covered by water surfaces and the upper threshold $h_{max}$ above which the pixel comprises only unponded sea ice surface. This threshold is a function of the temperature index $T_{idx}$. Inlay: The dependency of $h_{max}$ on the $T_{idx}$ derived from *in-situ* spectral albedo measurements. Displayed is a density plot of the *in-situ* data, and a linear fit (gray line) describing the threshold above which the surface is so bright that it is expected to be sea ice only, changing with $T_{idx}$.

Again, we make use of $T_{idx}$ to take care of the seasonal and regional differences, mainly driven by the air temperature
history. In this case, both thresholds are depending on $T_{idx}$, and additionally on the $twf$. This is because the $twf$ determines the fraction of the spectrum that is defined by the melt pond/open ocean mixture: If $twf$ is very small, the pixel spectrum will be dominated by the sea ice surface and have a small $s$, even though the small amount of water might be melt ponds with large $s$. Figure 12 displays the linear dependency of the upper and lower threshold $s_{min}$ and $s_{max}$ on $twf$. If there is almost no water in the pixel, the spectrum is fully dominated by the sea ice and it is impossible for the small amount of water to
distinguish between open ocean and melt ponds. This is one of the major issues of this empirical approach. However, it is a common problem to separate melt ponds and open ocean from moderate resolution spectroradiometers, like on Sentinel-3 or MODIS, in the presence of sea ice because of the spectral neutrality of open ocean (Rösel and Kaleschke, 2011; Istomina et al., 2015a, 2023). The value of $s$ in case of a pure sea ice surface is derived as a function of $T_{idx}$ from the *in-situ* spectral albedo measurements. This datasets provides spectra of sea ice surfaces only related to $T_{idx}$, from which the slope $s$ can be calculated.
The inlay in Figure 12 shows a density plot of this data with a linear fit of the average observed slope $s_{ice}(T_{idx})$. The other extreme case is the complete absence of sea ice surface in a pixel. This further separates into two possible extreme cases:
1. The sea ice is fully pond covered. For this case we found a threshold of $s_{pond} = 0.605$ to be suitable.
2. The pixel contains only open ocean. For this, a value of $s_{ocean} = 0.264$ proved to be suitable.



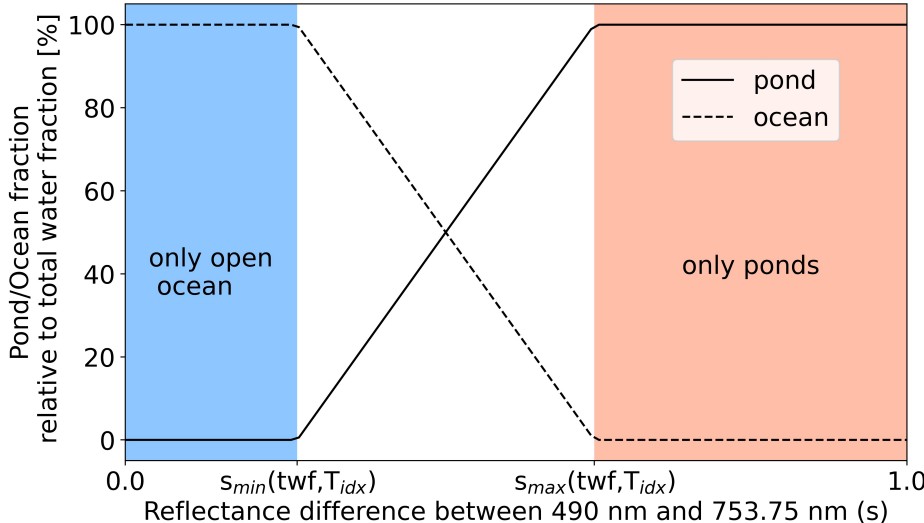

**Figure 11.** Pond fraction (solid line) and ocean fraction (dashed line) as a function of the spectrum slope $s$ and given relative to the total water fraction $twf$. Below the threshold $s_{min}$, all water in the pixel is open ocean. Above the threshold $s_{max}$, all water in the pixel is melt ponds. Both thresholds depend on $T_{idx}$ and $twf$.

These values are derived by testing combinations of these values in a reasonable range on the test pixels and then evaluating the results in comparison with the Sentinel-2 reference scenes. Note that these values relate to TOA reflectance observations, where atmospheric effect might lead to deviations from *in-situ* literature observations. Between the three tie points of extreme (pure surface type) cases, linear functions are used to describe the thresholds:

$$s_{min}(twf, T_{idx}) = s_{ice}(T_{idx}) + twf \cdot (s_{ocean} - s_{ice}(T_{idx})) \tag{22}$$

$$s_{max}(twf, T_{idx}) = s_{ice}(T_{idx}) + twf \cdot (s_{pond} - s_{ice}(T_{idx})) \tag{23}$$

### 5.3 Results of the empirical retrieval

Putting together the estimation of the total water fraction and the separation of melt ponds and open ocean, we derive the desired quantities:

$$f_{mp} = \frac{s_{mp}}{s_{ic}} = \frac{twf \cdot pf}{1 - twf \cdot of} \tag{24}$$

$$s_{oc} = twf \cdot of \tag{25}$$

With this, the estimate of $f_{mp}$ and $s_{oc}$ is improved from using constant values (Figure 13, upper row) to pixelwise adjusted values (Figure 13, middle row). Compared to the reference values from Sentinel-2 (Figure 13 bottom row), the uncertainty



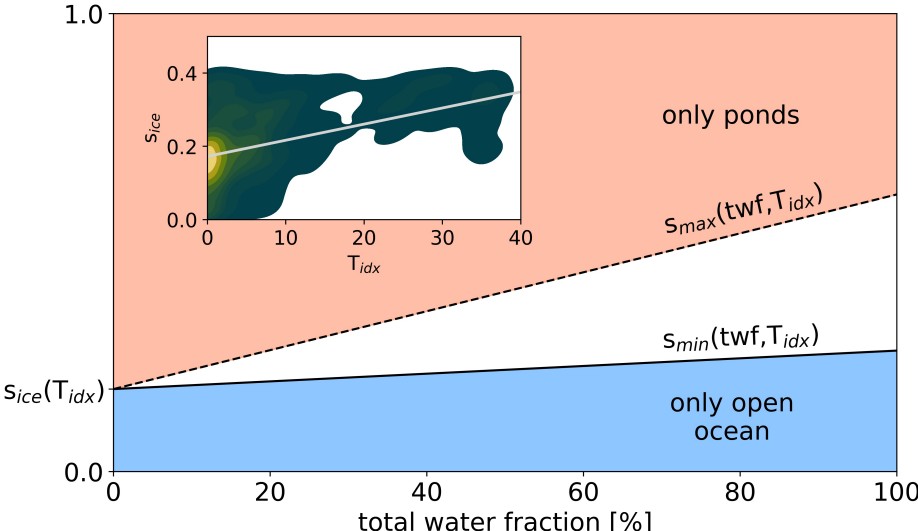

**Figure 12.** Main figure: lower threshold of the spectrum slope $s_{min}$ (solid line) below which all water is assumed to be open ocean and upper threshold $s_{max}$ (dashed line) above which all water is melt ponds as a function of $twf$. Inlay: The dependency of $s_{ice}$ on the $T_{idx}$ derived from *in-situ* spectral albedo measurements. Displayed is a density plot of $s$ derived from *in-situ* data, and a linear fit (gray line) of the average slope of the ice surfaces $s_{ice}$ in dependency on $T_{idx}$.

is reduced from above 20 % to approximately 10 % within the test pixels. This is expected to be sufficient to function as initialization for $f_{mp}$ and $s_{oc}$ for the MPD2 physical retrieval. Despite the lower uncertainty of the initialization values, the

boundaries are set looser to $\pm 25$ % of the initial values.





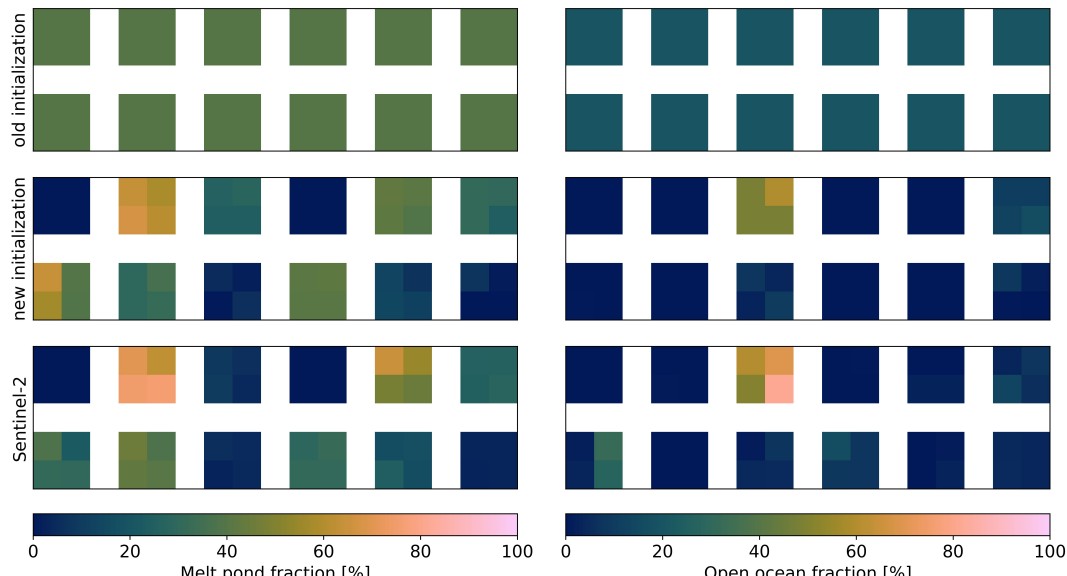

**Figure 13.** Melt pond fraction (left) and open ocean fraction (right) of the test pixels (see Figure 2). The upper row displays the previously used, constant initialization of 40 % and 20 % for $f_{mp}$ and $s_{oc}$, respectively. The middle row displays the new, pixel specific initialization. The lower row shows the reference from the Sentinel-2 data product.

# 6 Verification and discussion of results

## 6.1 Evaluation on test pixels

The new initialization of the four parameters $a_{eff}$, $\tau_{wi}$, $f_{mp}$ and $s_{oc}$ (Sections 4 and 5) is applied in another Monte Carlo simulation on the 48 test pixels introduced in Section 3.2. For the remaining parameters the same random seed is used as before, also the number of runs is the same. Based on these simulations, the optimal initial values and constraints for the remaining parameters can be established. These values are not adapted pixelwise but fixed as constants (Table 4 and are used for the entire Arctic and season. A table of all parameters with their initial values and boundaries is given in Table 4. To evaluate the improvement achieved by the newly developed pixel dependent initializations, we use again the definition of *reasonable* results with the thresholds as defined in Section 3.2. Figure 14 shows the percentage of the results that are *reasonable* with and without the new initializations. Additionally, the average deviation of $f_{mp}$ compared to the Sentinel-2 reference is shown.

By using the new initialization for the four parameters, the number of *reasonable* results has increased from an average of 30 % of the 30000 runs to an average of 81 %. The improvement is visible for all assigned surface type combinations as well as for the randomly chosen test pixels. Also, the deviation of $f_{mp}$ compared to the Sentinel-2 reference dataset is reduced in all cases. Especially for strongly ponded pixels the improvement is pronounced, while still featuring the largest deviations. On average, the deviation within these test pixels reduced from 14 % to 6 %. However, an improvement can be expected when looking only at the test pixels that were used to develop the new initializations and constraints. Therefore we will look at the



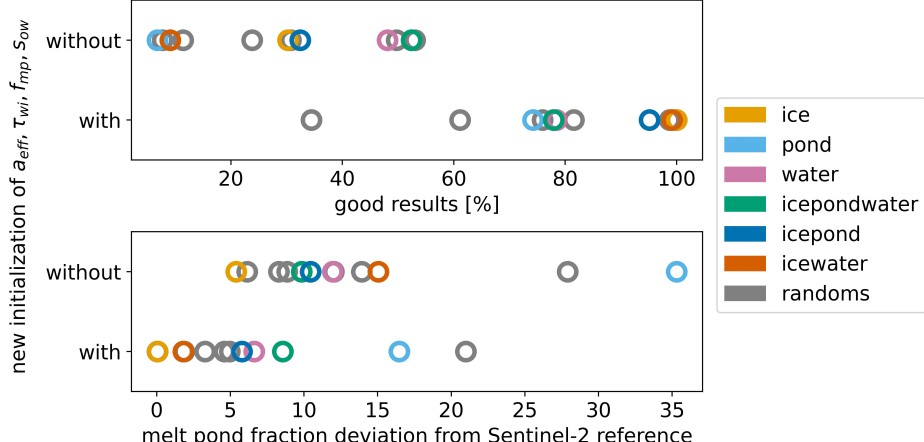

**Figure 14.** Comparison of the Monte Carlo simulation results with and without the new initilization of $a_{eff}$, $\tau_{wi}$, $f_{mp}$ and $s_{oc}$. The upper panel shows the percentage of the runs that gave a *reasonable* result. The lower panel shows average deviation of the retrieved $f_{mp}$ compared to the Sentinel-2 reference. The colors indicate the assigned and the random surface type pixels that were used in the simulations.

independent Sentinel-2 reference dataset next.

**Table 4.** Overview of the initial values, boundaries and increments of optimized parameters.

| Parameter | Symbol | Initial value | Lower boundary | Upper boundary | Increment |
|---|---|---|---|---|---|
| Melt pond fraction | $f_{mp}$ | pixelwise derived from reflectances | | | 0.0005 |
| Open water fraction | $s_{oc}$ | pixelwise derived from reflectances | | | 0.0005 |
| Effective grain size of ice surface | $a_{eff}$ | pixelwise derived from temperature history | | | $3\,\mu m$ |
| Optical thickness of white ice | $\tau_{wi}$ | pixelwise derived from temperature history | | | 0.1 |
| Absorption of yellow matter | $\alpha_y$ | $0.5\,\mathrm{m}^{-1}$ | $0\,\mathrm{m}^{-1}$ | $3\,\mathrm{m}^{-1}$ | $0.003\,^{-1}$ |
| Pond depth | $h_{pond}$ | $0.25\,\mathrm{m}$ | $0.0001\,\mathrm{m}$ | $4\,\mathrm{m}$ | $0.00001\,\mathrm{m}$ |
| Ice thickness beneath melt pond | $h_{ice}$ | $2\,\mathrm{m}$ | $0.1\,\mathrm{m}$ | $5\,\mathrm{m}$ | $0.01\,\mathrm{m}$ |
| Scattering coefficient of pond bottom | $\sigma_{ice}$ | $4\,\mathrm{m}^{-1}$ | $0.2\,\mathrm{m}^{-1}$ | $10\,\mathrm{m}^{-1}$ | $0.01\,\mathrm{m}^{-1}$ |

## 6.2 Comparison to Sentinel-2 reference

The new initilization is used for processing the Sentinel-3 data of all the occasions, where the Sentinel-2 reference dataset provides data. The possible time difference between the satellite overflights is regarded by the drift correction. Because of the time difference and varying cloud coverage some observations are discarded, if there is no overlapping data left. The same data is processed with the former version of the retrieval, i.e., MPD1. Figure 15 shows an example map of $f_{mp}$ derived with





MPD1 and MPD2 compared with the Sentinel-2 reference product. Figure 16 show $s_{oc}$ derived from MPD2 compared with
the Sentinel-2 product for the same example case.

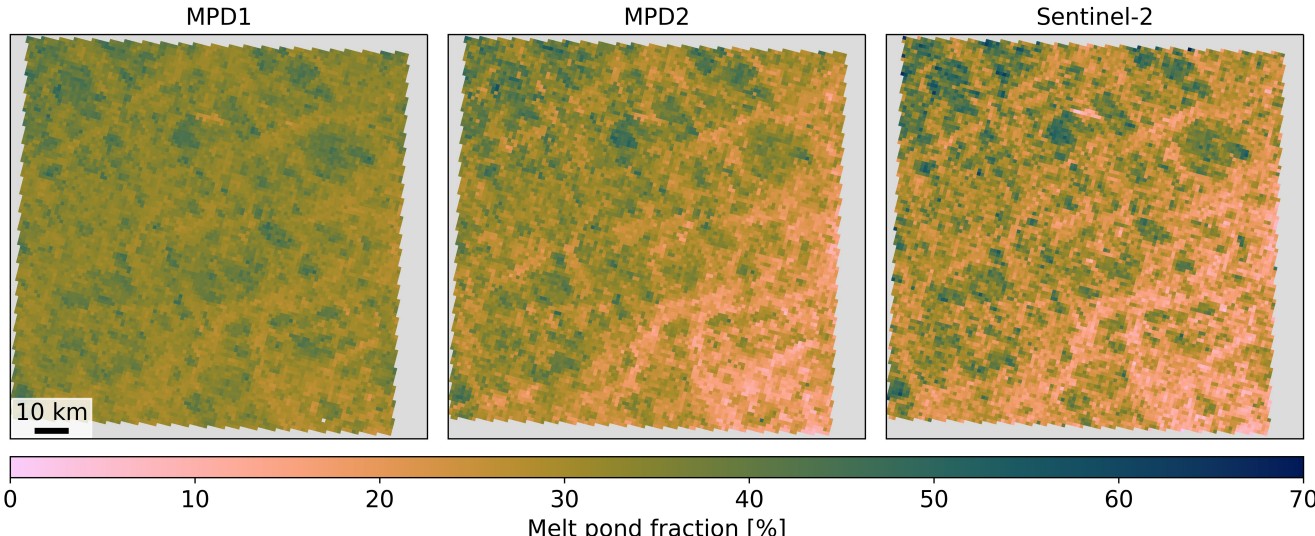

**Figure 15.** Regional maps of melt pond fraction $f_{mp}$ derived with the former algorithm MPD1 (left), the new retrieval version MPD2
(middle), and from the Sentinel-2 reference product (right). Shown is an example from July 3, 2017 close to the Queen Elizabeth Islands, at
approximately $79\,°$N and $120\,°$W. All maps share the same scale and color map.

The right panel of Figure 15 shows the reference $f_{mp}$ from Sentinel-2 scaled down to the $1.2\,$km resolution of the MPD
products. Despite the downscaling, it contains higher resolution information leading to greater detail. The left panel shows
the result from the MPD1 version. It is covering a much smaller range of $f_{mp}$ smoothing the details and missing the overall
tendency of higher $f_{mp}$ in the upper left trending towards lower values in the lower right. This trend is much better resolved
by the MPD2 version which is displayed in the middle panel of Figure 15.

$s_{oc}$ is only retrieved with the MPD2 algorithm and is not available for MPD1, thus only the MPD2 $s_{oc}$ can be compared to
sentinel-2 in Figure 16. The results for this case show consistently very low open ocean fractions. However, in the Sentinel-2
product few very small open ocean spots are detected which could not be resolved at the low resolution of the Sentinel-3 data
used in by the MPD2 algorithm.

Figure 17 shows the comparison of the retrieved MPD1 and MPD2 $f_{mp}$ with all Sentinel-2 reference scenes. The scatter
plot (left) shows the average values of the 33 scenes. While the MPD1 algorithm (cyan) struggles with low melt pond fractions,
MPD2 (dark blue) shows improved results with consistently low retrieved pond fractions. In general the results with MPD1
show a much smaller range of retrieved $f_{mp}$ than those from MPD2, which is in better agreement with the Sentinel-2 reference.
This is clearly displayed by the slopes of the linear regressions fitted to the data, which is increased from 0.49 to 0.82, while the
correlation coefficient of the linear regression remains almost the same. However, the coefficient of determination $R^2$ defined





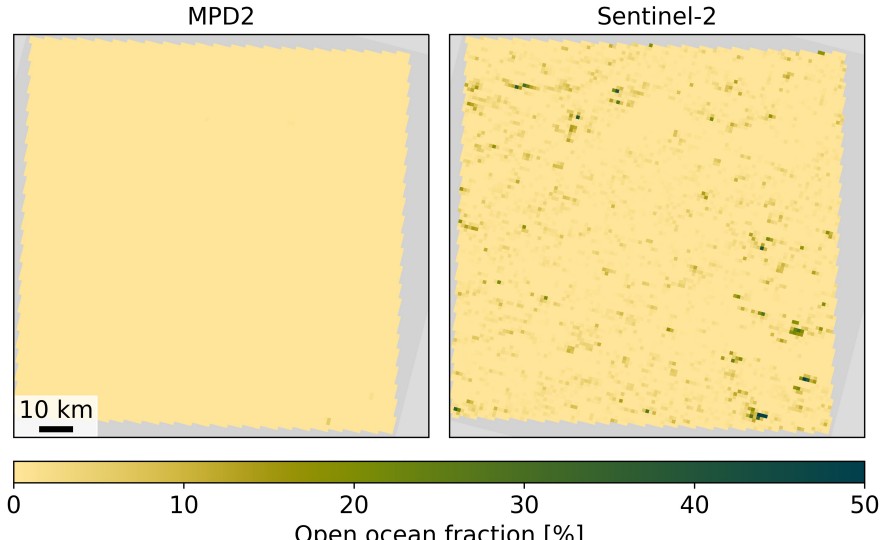

**Figure 16.** Regional maps of open ocean fraction $s_{oc}$ derived with the new algorithm version MPD2 (left) and from the Sentinel-2 reference product (right). Shown is an example from July 3, 2017 close to the Queen Elizabeth Islands, at approximately $79\,°$N and $120\,°$W. Both maps share the same scale and color map.

as

$$R^2 = \frac{\langle (f_{mp}^{MPD} - f_{mp}^{S2})^2 \rangle}{\langle (f_{mp}^{S2} - \overline{f_{mp}^{S2}})^2 \rangle} \qquad (26)$$

clearly emphasizes the improvement of the retrieval, as is increases from 0.39 for MPD1 to 0.89 for MPD2. The density plots (Figure 17 middle and right) show a comparison of the same data but without averaging all pixels of each scene. Again, the problem of retrieving low melt pond fraction values with MPD1 in these cases is clearly visible. The range of retrieved values for Sentinel-2 $f_{mp} < 5\%$ is very broad and barely reaches the range of the reference values. At the reference $f_{mp}$ close to $10\%$ the overestimation is the highest with an average of $17\%$. With increasing $f_{mp}$ the retrieval agrees better with the reference data. Overall, the majority of the reference data ranges from 0 to $35\%$ but is squeezed into a retrieved interval of 20 to $35\%$ by MPD1. This might be caused by the high initialization of $f_{mp} = 40\%$ for all pixels and the inability of the retrieval to escape a local minimum. Additionally, sub-pixel areas of open water often can be misclassified as melt ponds leading to the strong overestimation at low $f_{mp}$. This is also described by the bias of $+9.1$ describing a systematic tendency to overestimated $f_{mpf}$, while the RMSD is $12.9\%$. By using MPD2 with the additional open ocean class and the dynamic initialization of the state vector $\boldsymbol{X}$, the agreement is improved significantly. The uncertainty estimated from the RMSD compared to the reference data is reduced to $7.8\%$ and the bias is $+1.6$. This is comparable to uncertainties of other pan-Arctic melt pond fraction products by e.g., Lee et al. (2020); Ding et al. (2020), while adding understanding and usage of physical processes to the retrieval. For example, Rösel et al. (2012) present RMSD values in their comparison to reference data sets exceeding $10\%$. Peng et al. (2022) give an overview of different pan-Arctic melt pond fractions product, showing that the difference between the products





exceeds 5 % even when averaging over years, especially late in the melt season. Interestingly, two major modes of retrieved pond fractions are visible in MPD2: One at very low values, which is not surprising, as this relates to a completely unponded

surface which is naturally a common observation before the beginning of melt. The second mode is at $f_{mp} = 28\%$. This seems to be a common average value at the resolution of 1.2 km where extreme cases are rare because these would involve melt ponds with areas of around 1 km$^2$.

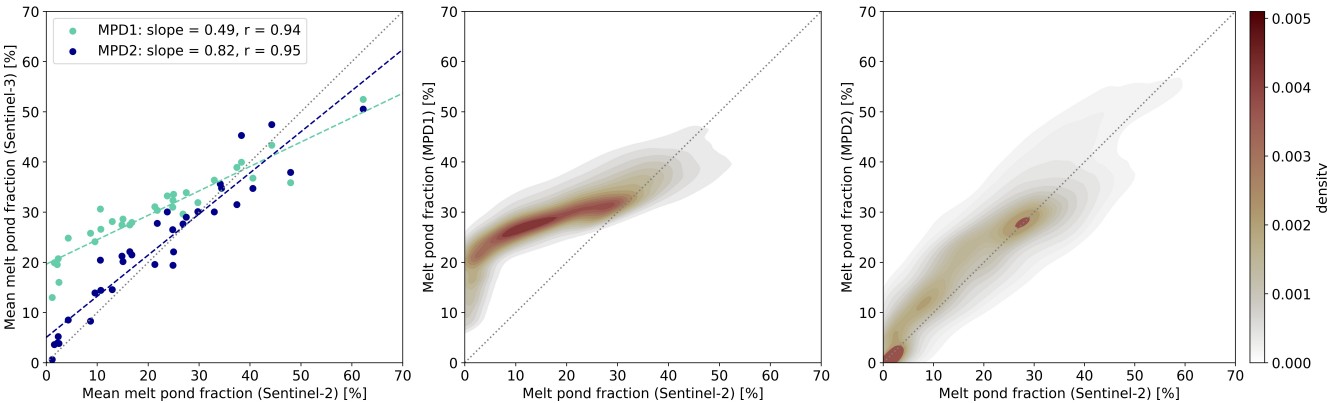

**Figure 17.** Left: Scatterplot of melt pond fraction $f_{mp}$ derived with MPD1 (cyan) and MPD2 (blue) against the Sentinel-2 reference product, averaged per Sentinel-2 scene. The dashed lines show a linear fit to the data points, the dotted line indicates the identity line. As a measure of quality, the slope and the correlation coefficient $r$ are specified. Middle and right: Density plots of $f_{mp}$ derived with MPD1 (middle) and MPD2 (right) against the Sentinel-2 reference product, compared pixelwise after drift correction. The color scale is normalized such that the area sums up to 1 and is valid for both densities.

The fraction of open ocean $s_{oc}$ newly retrieved with MPD2 is compared to the Sentinel-2 reference data in Figure 18. The scatter plot (Figure 18 left) of $s_{oc}$ shows a broader scattering than for $f_{mp}$ demonstrating the difficulties of detecting open ocean

due to its spectral neutrality in the visible and NIR range. Especially very small fractions <10 % are often not detected at all. This is most likely because there are not many cases with large open ocean areas such as polynyas but rather small leads and broken floes beyond the spatial resolution of the MPD. However, the slope of the linear regression is 0.83 and the correlation coefficient 0.79 showing reasonable agreement of the newly retrieved and the reference $s_{oc}$. The histogram of the pixelwise comparison of the same data in the right panel of Figure 18, shows that the agreement is almost perfect when there is no open

ocean at all in the pixel. That is the case, MPD1 was originally developed for. Once there is a little amount of open ocean within the pixel, MPD2 underestimates $s_{oc}$ in comparison to the Sentinel-2 reference. The strongest differences occur above open ocean fractions of 40 %. Up to 80 % $s_{oc}$, MPD2 is underestimating and then up to 95 % it is overestimating $s_{oc}$. The latter likely is because dark melt ponds (with very thin ice below) are being misclassified by the MPD2 algorithm because they already look almost alike open ocean. Overall the uncertainty of the open ocean product is estimated to 9.1 % (RMSD) with a

bias of $-0.8$.





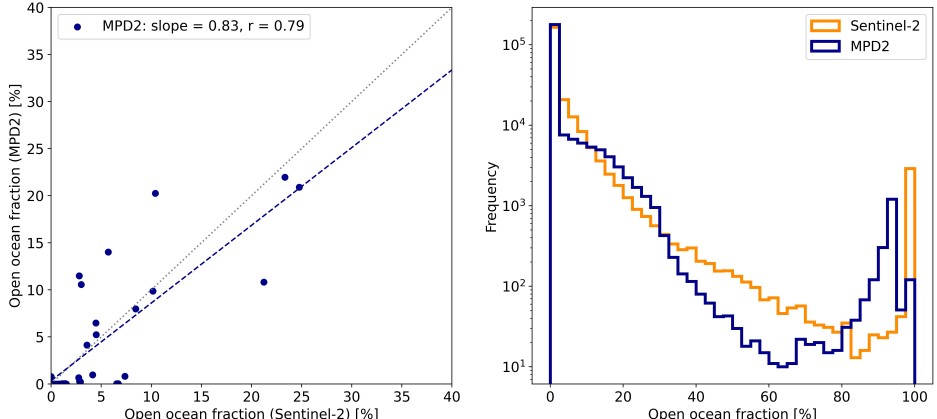

**Figure 18.** Left: Scatterplot of open ocean fraction $s_{oc}$ derived with MPD2 against the Sentinel-2 reference product, averaged per Sentinel-2 scene. The dashed line shows a linear fit to the data points. As a measure of quality, the slope and the correlation coefficient $r$ are specified. Right: Histograms of pixelwise $s_{oc}$ by Sentinel-2 (orange) and MPD2 (dark blue). Note the logarithmic scale of the y-axis.

### 6.3 Arctic-wide application of MPD2

Figure 19 shows an example of daily Arctic-wide maps of $f_{mp}$ (left), $s_{oc}$ (middle) and surface albedo produced with the new MPD2 algorithm from Sentinel-3 data. The swath-wise processed data is gridded into a polar stereographic grid (NSIDC grid) of 6.25 km resolution. In the course of this, the data is averaged daily, discarding grid cells containing less than 10 data points

or exceeding a standard deviation of the averaged values of 15 %. The broadband albedo displayed in the right panel of Figure 19, is derived from the retrieved spectral albedo with the spectral-to-broadband conversion developed by Pohl et al. (2020).

At the end of June, the melt season has already advanced. Especially on the level, landfast ice between the islands of the Canadian Archipelago melt pond fractions higher than 50 % are observed, which is in agreement with *in-situ* and higher resolution satellite observations from previous years (Landy et al., 2014; Li et al., 2020). Even at latitudes above 80°, melt

pond fractions of around 20 % are observed. This is not unusual for this time of the year as shown by results from the MOSAiC campaign in the same summer (Webster et al., 2022). The open ocean fraction is very low in the central Arctic, but closer to the sea ice edge, $s_{oc}$ reaches values beyond 20 %, indicating the break-up of ice floes with heterogeneous mixture of sea ice, melt ponds, and open ocean as the result. Thus, the broadband albedo is still quite high in the Central Arctic while already significantly reduced in the Fram Strait, Kara Sea, Laptev Sea, and Canadian Arctic Archipelago.

### 530 7 Conclusions

This study presents the approach of the new MPD2 algorithm to retrieve melt pond ($f_{mp}$) and open ocean ($s_{oc}$) fractions as well as the spectral and broadband albedo of sea ice from top of the atmosphere (TOA) reflectances in the optical and near infrared (NIR) range measured by satellite. This algorithm builds on the MPD1 algorithm developed by Zege et al. (2015) using



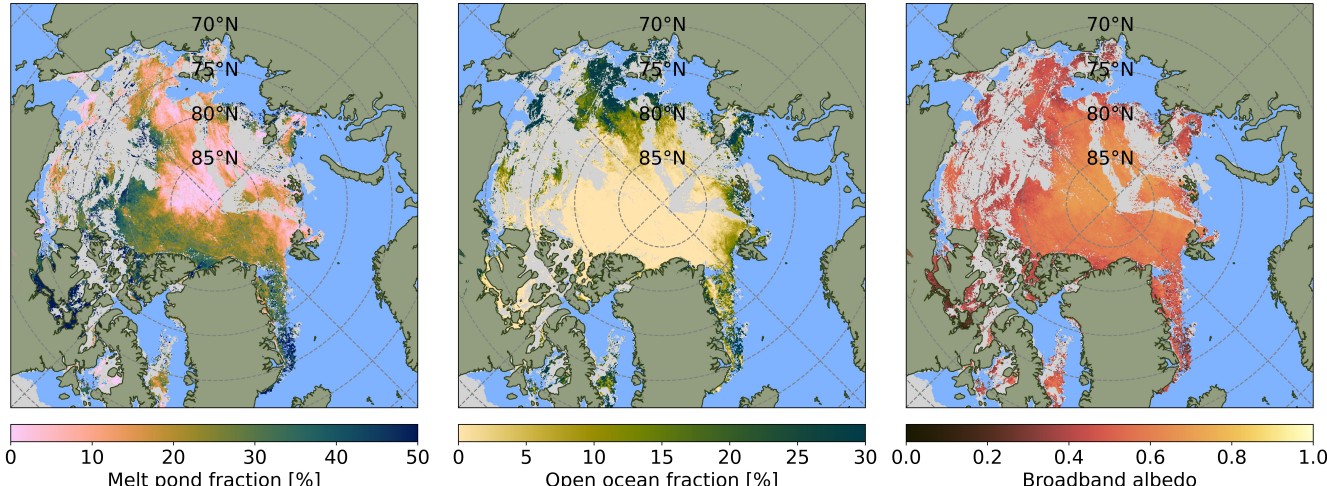

**Figure 19.** Pan-Arctic maps of melt pond fraction $f_{mp}$ (left), open ocean fraction $s_{oc}$ (middle), and broadband albedo (right) derived with MPD2 on June 30, 2020. The light blue color indicates open ocean where the retrieval has not been applied. Data gaps due to cloud contamination are gray.

the forward model by Malinka et al. (2016, 2018) to find an analytical solution for the bi-directional reflection at the sea ice surface. The novelty of the presented MPD2 algorithm is the introduction of the additional surface type class: open ocean. As shown in Istomina et al. (2023), moderate resolution spectroradiometers need additional data to retrieve 3 surface type classes as just TOA reflectances are not sufficient. To afford this, we have improved on the initialization of the optimized state vector $X$ describing the surface, thereby minimizing the risk of running into unreasonable local minima. 2-m air temperatures from ERA5 are used to calculate a temperature index along the drift track of the sea ice which is tracked with OSI-SAF sea ice drift data (Ocean and Facility). With this temperature index we account for regional and seasonal changes, relating to the physical properties of the sea ice surface. In this way the effective grain size $a_{eff}$ and the optical thickness $\tau_{wi}$ of the ice surface layer are constrained. Without such constrains the spectral unmixing on sub-pixel satellite scale of the three surface types - ice, melt ponds, and open ocean - is not reliably possible. Additionally, we make use of the different spectral behavior of these three surface types (Tschudi et al., 2008; Rösel et al., 2012; Istomina et al., 2015a; Light et al., 2022) to initialize and constrain their area fractions. In fact, a first order, empirical retrieval of $f_{mp}$ and $s_{oc}$ from TOA reflectances in the range 400 nm to 900 nm has been developed and is implemented as initial guess for the physical forward model.

By using the previously constructed temperature index, seasonal and regional differences of the sea ice optical properties defining its spectral behavior are taken into account. With these improvements, the uncertainty of $f_{mp}$ could be reduced from 12.9 % (MPD1) to 7.8 %, while the uncertainty of $s_{oc}$ is estimated to be 9 %. Furthermore, the bias of overestimating $f_{mp}$ has been significantly reduced to $+1.7$ and the coefficient of determination compared to the reference Sentinel-2 data set has been increased from 0.39 to 0.89. This is comparable to uncertainties of other melt pond fraction retrievals, e.g. by Rösel et al. (2012); Ding et al. (2020), and lower than the overestimation bias previously reported (Wright and Polashenski, 2020). While



other pan-Arctic melt pond fraction retrievals use artificial neural networks, we have presented a fully physical algorithm to detect melt ponds on sea ice and additionally also the fraction of open ocean in the Arctic summer. The longer time series

of melt pond and open ocean fractions from Sentinel-3 satellite data processed with MPD2 will be made available as netcdf files via https://seaice.uni-bremen.de/. In future, it is also possible to apply the algorithm to ENVISAT data and thereby extend the time series back to 2002. This can be very useful for, e.g., the study of the sea ice energy budget (Perovich et al., 2002; Nicolaus et al., 2012; Katlein et al., 2021) in the summer period. Thus it provides great potential to improve global climate models Hunke et al. (2013); Dorn et al. (2018) and better understand climate changes in the Arctic.

*Data availability.* The Sentinel-3 satellite data used, is publicly available under https://ladsweb.modaps.eosdis.nasa.gov/archive/allData/450/ (last access August 14, 2023).

The Sentinel-2 melt pond fraction product is available at PANGAEA: https://doi.org/10.1594/PANGAEA.950885.

ERA5 data are made available by the Copernicus Climate Change Service (C3S) at https://cds.climate.copernicus.eu/cdsapp#!/dataset/reanalysis-era5-single-levels?tab=form.

The OSI-405-c sea ice drift data is available via https://thredds.met.no/thredds/catalog/osisaf/met.no/ice/drift_lr/merged/catalog.html.

The spectral albedo data used, is publicly available on PANGAEA and data.meereisportal.de and can be retrieved at the following list of links: TARA albedo measurements: https://doi.org/10.1594/PANGAEA.945286. Spectral albedo during Alert MAPLI18 measurement campaign: https://doi.pangaea.de/10.1594/PANGAEA.949614. Spectral albedo from PS106-ARK31/1 expedition: https://data.meereisportal.de/relaunch/buoy.php?buoytype=RB®ion=all&buoystate=all&expedition=all&submit1=Anzeigen&active-tab1=method&ice-type=buoy&lang=

de&timeline=buoy&active-tab2=buoy&showMaps=y&dateRepeat=n. Albedo measurements from the MOSAiC campaign for the different stations: https://doi.pangaea.de/10.1594/PANGAEA.948876, https://doi.pangaea.de/10.1594/PANGAEA.948828, https://doi.pangaea.de/10.1594/PANGAEA.948712, https://doi.pangaea.de/10.1594/PANGAEA.948572.

*Competing interests.* The authors declare that they have no conflicts of interest.

*Acknowledgements.* We gratefully acknowledge the funding by the Deutsche Forschungsgemeinschaft (DFG, German Research Foundation) through the Transregional Collaborative Research Centre TRR-172 "ArctiC Amplification: Climate Relevant Atmospheric and SurfaCe Processes, and Feedback Mechanisms (AC)3" (grant 268020496) and the European Union's Horizon 2020 project CRiceS (grant 101003826). LI was funded through the EU Horizon 2020 project SPICES (grant 640161), the project REASSESS (grant 424326801) and the DFG SPP 1158 "Antarctic Research" project. MN was partly funded through the EU Horizon 2020 project Arctic Passion (grant 101003472). Part of

the data used in this article were produced as part of the international MOSAiC project with the tag MOSAiC20192020 and the Project_ID:



AWI_PS122_00. We thank all people involved in the expedition of the R/V Polarstern during MOSAiC in 2019–2020 as listed in Nixdorf et al. (2021).



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
