# Peer review of "Melt pond fractions on Arctic summer sea ice retrieved from Sentinel-3 satellite data with a constrained physical forward model"

_EGUsphere, 2023_

## Author Comment (AC1)

**Melt pond fractions on Arctic summer sea ice retrieved from Sentinel-3 satellite data with a constrained physical forward model**

**Response to reviewer 1**

We thank the reviewer very much for the time and effort to thoroughly read and evaluate our manuscript to improve its quality.

Overall:  Melt ponds play a critical role in assessing the reflectance of sea ice and thus the surface energy balance.  Determination of areal melt pond coverage is challenging, due to the variability in individual pond reflectance, and to the flat spectral reflectance of snow-covered ice and open ocean.  The MDP2 technique presented in this paper utilizes physical parameterizations to determine individual pond reflectance rather than a single fixed reflectance curve, as well as 4 surface types to more accurately estimate pond coverage from a high-resolution satellite-based sensor (Sentinel-2).  The results presented show promise to improve pond coverage parameterizations in sea ice models, as well as regional studies of the evolution of the sea ice surface.

The MDP2 technique is sound and presented clearly.  The paper is well-written and the figures are important to explain intermediate and final results.

I recommend accepting this paper after addressing a couple comments and the corrections that follow:

Thank you very much for your positive and encouraging feedback. In the following we will address every comment separately.

Specific comments:

Line #    Comment

95    Should provide a citation for OSI-SAF drift data, and mention its tracking error

We will add a reference to the download source of the OSI-SAF drift product. Also the product comes along with an uncertainty estimate of the drift  for every single data point. We will add this comment as well as a rough estimate of the average uncertainty to the data description section.

124    should read "from 2012 to 2022"

Thanks, we correct the order of the years.

166     The assumption of clear melt water may lead to occasional error in actual pond reflectance, with dust and other aerosols present in or near the bottom of the pond.  This is particularly true near coastal areas but have been shown to be transported from regions with high industrial activity.  This would be difficult to address in this pond estimation technique but should be mentioned as a possible impact on pond reflectance.

The used model of melt pond reflectance was carefully verified with field measurements in (Malinka et a; 2018). Most of the melt ponds observed on Arctic sea ice in three different expeditions were quite clear and showed very good coincidence of the measured and modeled spectra. Some melt ponds were polluted and showed a decrease in spectral albedo in the blue region by a maximum value of 0.03 with an rmsd = 0.1. However, the ponds of these kind were rather dark and had an albedo below 0.3 in the blue range. As it is the reduction of albedo that determines the actual retrieved pond fraction, the presence of a sediment will result in an overestimation of the melt pond fraction by (1-0.3)/(1-0.3-0.03), i.e. 4% as an upper bound, and (1-0.3)/(1-0.3-0.01), i.e. 1.4% as an rmsd assessment.

We will mention in the manuscript that close to the coast this might not necessarily 100% correct but will not effect the retrieval outcome significantly:

"Although this might not be perfectly true, especially in near coastal areas, the effect on the retrieval results in the form of an overestimation of melt pond fraction is estimated to be negligibly small (Malinka et al. 2018)."

**230    Should read "This additional information…"**

We correct that.

**286    Should read "no longer considered…"**

We correct that.

**297    Should read "are NOT appropriate…"**

We correct that.

**372    Should read "both differences…"**

Thanks for spotting.

**456    End of sentence continues after Fig 14 – hard to find**

We will take care of this during the final editing process if this is still a problem then.

**525    Does Webster or others have MOSAiC aerial melt pond coverage, that can be directly compared to the derived pond fraction from MPD2?**

Airborne melt pond fraction observations usually only cover small parts of the Sentinel-3 footprints. Because of the spatial variability of melt pond fraction this would lead to inaccurate comparisons which are hard to interpret. That is the reason why we decided to use the hierarchy airborne – Sentinel-2 – Sentinel-3 to compare and evaluate our product. In our preceding paper (Niehaus et al. 2023, https://doi.org/10.1029/2022GL102102) we have compared the Sentinel-2 melt pond fraction to airborne data.

**559    need brackets about citations**

We correct that.

---

## Author Comment (AC2)

**Melt pond fractions on Arctic summer sea ice retrieved from Sentinel-3 satellite data with a constrained physical forward model**

**Response to reviewer 2**

We thank the reviewer very much for the time and effort to thoroughly read and evaluate our manuscript to improve its quality.

The authors build on previous work on deriving melt pond fraction from Sentinel-3. The main improvement is including open ocean as additional surface type classification. The improvements to the algorithm are impressive when compared to the old algorithm and a better pan arctic melt pond fraction data is really good news! The methodology is sound and detailed, and the authors have done a good job of explaining it all in a logical order. Additionally, the figures are clear and contain a lot of information. I would have liked to see a more thorough assessment of errors (i.e. do they vary based on melt pond fraction), but given the lack of data for the and length of the paper it would be good to see it in future work.

I recommend accepting this paper.

Thank you very much for your positive and encouraging feedback. In the following we will address every comment separately.

Above you mention your interest in a more detailed discussion of errors. We looked into the dependence of the error (i.e. the difference between retrieval output and validation data) on the value of melt pond fraction itself and don't observe a dependence. We will add this information to the manuscript where the uncertainties are discussed.

L43: Satellites don't cover all of the arctic.

We don't claim that all satellites do cover all of the Arctic. However, we say that they can and this is the case for the Sentinel-3 satellites which don't have a pole hole.

L175: ?

Thank you for spotting this wrong citation key which led to the ?. We will correct that.

L373: bothdifferences

We correct this.

In Figure 15, it would be nice to see a few more of the example scenes (and perhaps the S2 image?)

We will add the comparison shown in Figure 15 for some other example scenes to the appendix.